# Tuning of in vivo cognate B-T cell interactions by Intersectin 2 is required for effective anti-viral B cell immunity

Marianne Burbage[1][†]*, Francesca Gasparrini[1], Shweta Aggarwal[1], Mauro Gaya[1,2], Johan Arnold[2], Usha Nair[2], Michael Way[3], Andreas Bruckbauer[1][‡], Facundo D Batista[1,2]*

[1]Lymphocyte Biology Laboratory, The Francis Crick Institute, London, United Kingdom; [2]Ragon Institute of MGH, MIT and Harvard, Cambridge, United States; [3]Cellular Signalling and Cytoskeletal Function Laboratory, The Francis Crick Institute, London, United Kingdom

*For correspondence:
marianne.burbage@curie.fr (MB);
FBATISTA1@mgh.harvard.edu
(FDB)

Present address: [†]Institut Curie, Paris, France; [‡]FILM facility, Imperial College London, London, United Kingdom

**Abstract** Wiskott-Aldrich syndrome (WAS) is an immune pathology associated with mutations in WAS protein (WASp) or in WASp interacting protein (WIP). Together with the small GTPase Cdc42 and other effectors, these proteins participate in the remodelling of the actin network downstream of BCR engagement. Here we show that mice lacking the adaptor protein ITSN2, a G-nucleotide exchange factor (GEF) for Cdc42 that also interacts with WASp and WIP, exhibited increased mortality during primary infection, incomplete protection after Flu vaccination, reduced germinal centre formation and impaired antibody responses to vaccination. These defects were found, at least in part, to be intrinsic to the B cell compartment. In vivo, ITSN2 deficient B cells show a reduction in the expression of SLAM, CD84 or ICOSL that correlates with a diminished ability to form long term conjugates with T cells, to proliferate in vivo, and to differentiate into germinal centre cells. In conclusion, our study not only revealed a key role for ITSN2 as an important regulator of adaptive immune-response during vaccination and viral infection but it is also likely to contribute to a better understanding of human immune pathologies.
DOI: https://doi.org/10.7554/eLife.26556.001

## Introduction

B lymphocytes play an integral part in humoral immunity through their ability to produce high affinity antibodies in response to infections. Naïve B cells express a unique B cell receptor (BCR) that allows them to recognise cognate antigen, usually on the surface of antigen presenting cells (APCs), such as follicular dendritic cells or subcapsular sinus macrophages (*Batista and Harwood, 2009*; *Cyster, 2010*). Binding of antigen to the BCR triggers a signaling cascade, which provides the first signal for B cell activation (*Kurosaki et al., 2010*). BCR engagement also induces rapid and radical alterations of cellular morphology, coupled to substantial remodelling of the B cell actin network (*Fleire et al., 2006*).

BCR-induced reorganisation of the actin cytoskeleton plays a central part in the initiation of B cell responses. Consequently, B cells lacking key cytoskeleton regulators, such as the small Rho GTPases Rac1 and Rac2, RhoA or Cdc42, show far-reaching functional impairments (*Guo et al., 2009*; *Burbage et al., 2015*; *Gerasimcik et al., 2015*; *Saci and Carpenter, 2005*; *Walmsley et al., 2003*). The activity of Rho GTPases is modulated by G-nucleotide exchange factors (GEFs) that trigger their switch from a GDP-bound inactive to a GTP-bound active state. B cell-intrinsic deletion of the GEFs Vav or dedicator of cytokinesis (DOCK8) severely impairs antibody responses in mice (*Fujikawa et al., 2003*; *Randall et al., 2009*; *Tarakhovsky et al., 1995*). Accordingly, in humans,

multiple immune pathologies have been linked to mutations in actin regulators. Mutations in *DOCK8* have been found in independent cohorts of immunodeficient patients (*McGhee and Chatila, 2010*; *Zhang et al., 2009*). Wiskott-Aldrich syndrome (WAS), characterised by recurrent infections and abnormal lymphocyte function is commonly caused by loss-of-function mutations in WAS protein (WASp) or in its interacting protein WIP (*Lanzi et al., 2012*; *Thrasher and Burns, 2010*), both of which are involved in triggering actin polymerisation downstream of Cdc42 (*Martinez-Quiles et al., 2001*; *Moreau et al., 2000*).

One consequence of BCR signalling is antigen internalisation followed by its processing and presentation onto MHC class II, enabling cognate interactions between activated B cells and CD4 T lymphocytes that recognise antigenic peptide-MHC complexes (*Lanzavecchia, 1985*). These interactions allow B cells to receive T cell help in a contact dependent fashion. The combination of BCR signalling and T cell help is critical for B cells to enter the germinal centre (GC) reaction, during which they undergo somatic hypermutation and class-switch recombination, and from where antibody secreting cells with high affinity for the antigen emerge (*Victora and Mesin, 2014*). The establishment of prolonged contacts between B and T cells rely on interactions between various receptors, such as MHCII and TCR, or CD80/CD86 and CD28 (*Crotty, 2015*). The signalling lymphocytic activation molecule (SLAM) family of transmembrane receptors and the SLAM-associated protein (SAP) family of intracellular adaptors have crucial roles in stabilising B-T conjugates both at the B-T border and in GCs (*Schwartzberg et al., 2009*). In humans, mutations in *SH2D1A* (encoding for SAP) are causative of X-linked lymphoproliferative disease (XLP), an immune pathology characterised by an extreme sensitivity to Epstein Barr Virus infections (*Coffey et al., 1998*; *Sayos et al., 1998*). SAP deficient T cells fail to establish long term conjugates with B cells, resulting in a dramatic block in the generation of high affinity antibodies (*Crotty et al., 2003*; *Czar et al., 2001*; *Qi et al., 2008*). The SLAM receptor CD84 is also required for this process, as CD84 deficient T cells are compromised in their ability to establish stable interactions with B cells (*Cannons et al., 2010*). However, the molecules involved in stabilising these contacts on the B cell side have not been identified.

Intersectin (ITSN) 2 is a multimodular adaptor protein ubiquitously expressed as two distinct isoforms, ITSN2-S, and the longer isoform ITSN2-L. Both isoforms contain SH3 domains, which support direct interactions with multiple partners, including WASp and WIP (*Gryaznova et al., 2015*), but only ITSN2-L has a GEF domain consistent of a DH (Dbl homology) domain coupled to a PH (Pleckstrin homology) domain (*Pucharcos et al., 2000*). Notably, the GEF activity of ITSN2-L is selective for Cdc42 (*Novokhatska et al., 2011*). Interestingly, in a large-scale association study, *ITSN2* has been identified as a potential at-risk locus for Sjögren's syndrome, a common autoimmune pathology characterised by keratoconjunctivitis and xerostomia (*Lessard et al., 2013*). Moreover, the *ITSN2* locus has been found to be differentially methylated in B lymphocytes from healthy donors versus cells from Sjögren's syndrome patients (*Miceli-Richard et al., 2016*).

In this study, we provide the first characterisation of the role of ITSN2 in the context of immune responses. We show that genetic ablation of ITSN2 rendered mice more sensitive to a lethal infection with Influenza virus. Furthermore, ITSN2 deficient B cells were defective in entering the GC reaction and in generating high affinity antibodies. In vivo, *Itsn2^{-/-}* B cells exhibited proliferation defects upon immunisation, expressed reduced levels of various surface receptors, and were impaired in forming long-term conjugates with cognate T lymphocytes. The results presented here provide the first characterisation of the role of ITSN2 in the context of immune responses. Furthermore, they identify an essential function for this protein in the regulation of B-T cell interactions, germinal centre formation and antibody production, which is reminiscent of the phenotype associated with SAP or CD84 deficiency in T cells.

## Results

### B and T cells develop normally in *Itsn2^{-/-}* mice

Due to the intricate relationship between BCR signalling, the actin cytoskeleton and its regulators, we sought to characterize the role of ITSN2 in mouse immune responses. To analyse the function of ITSN2 in B cells, we obtained ITSN2 deficient mice from the Knockout Mouse Project (KOMP) consortium. These animals were generated using the Velocigene technology; they carry a LacZ reporter cassette knocked into the *Itsn2* locus, disrupting the expression of this gene, and a selectable

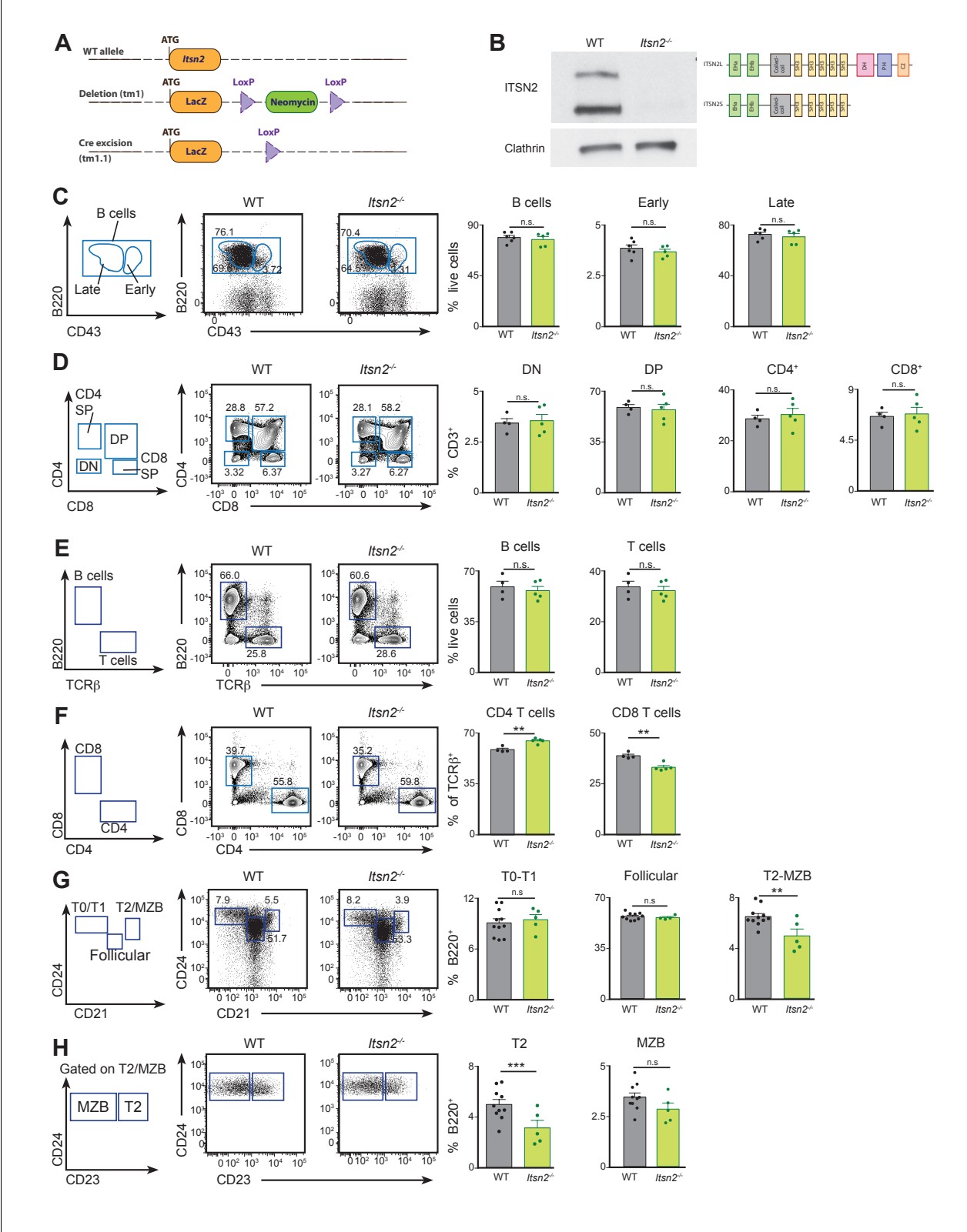

**Figure 1.** Lymphocyte development is not compromised by ITSN2 deletion. (**A**) Genetic approach used to delete ITSN2. A LacZ cassette was inserted in the *Itsn2* locus to disrupt protein expression. A neomycin resistance cassette flanked by two loxP sites was used as a selection marker, and subsequently excised by Cre-mediated recombination (KOMP allele tm1.1). (**B**) Naïve B cells were purified from the spleens of WT and *Itsn2*−/− mice and protein expression of ITSN2 (upper immunoblot, two isoforms as illustrated on the right) and clathrin heavy chain (lower immunoblot) were detected by

*Figure 1 continued*

Western blot. (C) Bone marrow from WT and *Itsn2^-/-* littermates was analysed by flow cytometry using the gating strategy shown on the left. B cell progenitors (B220$^+$) were divided into immature (CD43$^+$) and mature (CD43$^-$) cells on the basis of CD43 expression. Data quantified in the panels on the right show the percentage of live cells in the indicated gates. (D) Thymus from WT and *Itsn2^-/-* littermates were analysed by flow cytometry. Gating strategy is shown on the left. T cell progenitors (CD3$^+$) were divided into double negative (CD4$^-$ CD8$^-$), double positive (CD4$^+$ CD8$^+$), CD4 single positive (CD4$^+$ CD8$^-$) and CD4 single positive (CD4$^-$ CD8$^+$) cells. Data quantified in the panels on the right show the percentage of CD3$^+$ cells in the indicated gates. (E–H) Spleens from WT and *Itsn2^-/-* were analysed by flow cytometry. (E) Identification and quantification of B cells (B220$^+$ TCRβ$^-$) and T cells (B220$^-$ TCRβ$^+$). (F) T cells were subdivided into CD4 (CD4$^+$ CD8$^-$) and CD8 (CD4$^-$ CD8$^+$) T cells and quantified. (G–H) B cells were divided on the basis of CD21, CD23 and CD24 expression into T0/T1 cells (CD21$^-$CD24$^{hi}$, G), follicular B cells (CD21$^{lo}$CD24$^{lo}$, G), T2 cells (CD21$^{hi}$ CD24$^{hi}$ CD23$^-$, H), MZB cells (CD21$^{hi}$ CD24$^{hi}$ CD23$^+$, H) and quantified (right panels). For all flow cytometry experiments (C–G), data are from 1 out of 3 representative experiment with 4 or more animals in each group, and each dot represents an individual mouse. Student's t-test, ns p>0.05, *p<0.05, **p<0.01, ***p<0.001.

DOI: https://doi.org/10.7554/eLife.26556.002

The following figure supplement is available for figure 1:

**Figure supplement 1.** Lymphocyte development is not compromised by ITSN2 deletion.

DOI: https://doi.org/10.7554/eLife.26556.003

neomycin marker that was subsequently be excised by Cre recombinase (*Figure 1A*, [*Skarnes et al., 2011*; *Valenzuela et al., 2003*]). ITSN2 is a multimodular adaptor protein with two alternative stop codons yielding functionally distinct isoforms, ITSN2-L and ITSN2-S, with only ITSN2-L bearing a GEF domain (DH-PH) (*Pucharcos et al., 2000*). While we could detect the expression of both ITSN2 isoforms in wild type (WT) B cells, this expression was abolished in B cells from the ITSN2 knockout (Itsn2tm1.1(KOMP)Vlcg) animals, hereafter referred to as *Itsn2^-/-* (*Figure 1B*).

To examine the effect of ITSN2 deletion on early lymphocyte development, we extracted bone marrow (*Figure 1C*) or thymus (*Figure 1D*) cells from 8 to 12 week-old WT and *Itsn2^-/-* mice and analysed them by flow cytometry. B cell progenitors were analysed using the Hardy classification system (*Hardy et al., 1991*). Bone marrow cellularity was largely unaltered by ITSN2 deletion, and we found similar numbers of early (B220$^+$CD43$^+$) and late (B220$^+$CD43$^-$) B cell progenitors in bone marrows from WT and *Itsn2^-/-* mice (*Figure 1C*). Moreover, we did not detect any differences in frequencies of the Hardy populations A (CD43$^-$ CD24$^-$ BP-1$^-$), B (CD43$^{hi}$ CD24$^+$ BP-1$^-$), C (CD43$^-$ CD24$^+$ BP-1$^+$), D (CD43$^-$ IgM$^-$ IgD$^-$), E (CD43$^-$ IgM$^+$ IgD$^-$) and F (CD43$^-$ IgM$^+$ IgD$^{hi}$) cells upon ITSN2 deletion (*Figure 1—figure supplement 1A and B*). This suggests that lack of ITSN2 does not compromise the ability of B cells to undergo rearrangements of the BCR heavy and light chain loci (stages A-B and D, respectively) and subsequent positive selection and proliferation (stage C and E, respectively). In the thymus, T cell progenitors (CD3$^+$) are subdivided on the basis of CD4 and CD8 expression into double negative (CD4$^-$CD8$^-$), double positive (CD4$^+$CD8$^+$), CD4 single positive (CD4$^+$CD8$^-$) and CD8 single positive (CD4$^-$CD8$^+$). These four populations were found in comparable numbers in WT and *Itsn2^-/-* animals. (*Figure 1D*). These results indicate that ITSN2 is not required for early lymphoid development.

To examine the impact of ITSN2 deletion on lymphocyte populations on secondary lymphoid organs, we analysed the spleens of 8–10 week old WT and *Itsn2^-/-* littermates by flow cytometry. We detected comparable numbers of B and T cells in animals from both genotypes (*Figure 1E*). However, we detected a small increase in the fraction of CD4 T cells mirrored by a decrease in that of CD8 T cells in *Itsn2^-/-* mice compared to WT animals (*Figure 1F*).

To characterise the progression of transitional B cells towards fully mature B lymphocytes, we analysed single cell suspensions from the spleens of WT and *Itsn2^-/-* mice by flow cytometry. Within the B cell compartment, we found comparable proportions of T1 cells (B220$^+$CD21$^{lo}$CD24$^{hi}$), suggesting that homing of immature B cells to the spleen is not compromised by ITSN2 deletion. In contrast, there was a 50% reduction in numbers of T2 (B220$^+$CD21$^{hi}$CD24$^{hi}$ CD23$^+$) B cells upon ITSN2 deletion (*Figure 1G*). As the survival of T2 cells is highly dependent on BCR and BAFF-mediated signalling, these results suggest that *Itsn2^-/-* T2 B cells may have a defect in sensing these signals (*Hsu et al., 2002*; *Schiemann et al., 2001*). However, *Itsn2^-/-* B cells retained some maturation potential, as we found similar amounts of both mature follicular (B220$^+$CD21$^{hi}$CD24$^{lo}$) and marginal zone B cells (B220$^+$CD21$^{hi}$CD24$^{hi}$CD23$^-$) in WT and *Itsn2^-/-* mice (*Figure 1G and H*). Moreover, WT and *Itsn2^-/-* animals had comparable numbers of B2 (CD5$^-$CD11b$^-$), B1a (CD5$^+$CD11b$^+$) and B1b

(CD5⁻CD11b⁺) cells in the peritoneal cavity (*Figure 1—figure supplement 1C*). Concordant with these results, basal titres of IgM, IgG1, IgG2b and IgG2c were comparable in WT and *Itsn2⁻/⁻* animals (*Figure 1—figure supplement 1D*). Together, these results suggest that ITSN2 is broadly dispensable for the establishment of mature lymphocyte populations.

## Higher susceptibility to viral infection and diminished germinal centre formation in *Itsn2⁻/⁻* mice

In order to determine the physiological consequences of ITSN2 deletion, we intranasally infected 12–16 weeks WT and *Itsn2⁻/⁻* littermates with a lethal dose ($3 \times 10^4$ PFU) of the Influenza A virus (PR8 strain). Post-infection, we followed the survival of these animals by weighing them daily and sacrificing the ones that exhibited 20% weight loss. We found that 4 and 5 days after viral infection, *Itsn2⁻/⁻* mice showed significantly lower body weight than their WT counterparts (*Figure 2B*). As a result, the survival of the ITSN2 KO animals was reduced (p=0.0345) compared to that of WT littermates upon influenza infection (*Figure 2A*). Indeed, the first *Itsn2⁻/⁻* animals were sacrificed as early as 3 days post infection, while the majority of WT animals survived until day six post-infection. These results suggest that ITSN2 is required for mounting a robust response to viral infection.

To further define the role of ITSN2 in the establishment of protective responses, we immunised WT and *Itsn2⁻/⁻* mice with hemagglutinin trimer (*Koday et al., 2016*; *Zheng et al., 2016*). After one month, animals were challenged with a lethal dose ($1.5 \times 10^5$) of Influenza PR8. Upon such challenge, non-immunised WT animals lost weight rapidly, and had to be sacrificed within 1 week of infection (data not shown). Immunised WT animals also exhibited rapid weight loss, but started to regain weight 9 days after infection. In contrast, immunised *Itsn2⁻/⁻* animals lost weight at similar rate as their WT counterparts, and their weight remained low until d14 (*Figure 2C*). These results indicate that vaccinated *Itsn2⁻/⁻* animals fail to control the infection compared to WT vaccinated mice. Hence, these results support the notion that ITSN2 is required for establishing protective immune responses against viral infection.

To gain insights into which cellular processes underlying the humoral immune response were impaired in *Itsn2⁻/⁻* mice, we infected WT and *Itsn2⁻/⁻* animals with a sub-lethal dose of $10^4$ PFU Vaccinia virus (Western reserve) via intra-footpad injection and analysed the popliteal lymph nodes by flow cytometry 7 days after infection. We found similar numbers of CD19⁺ B cells in animals from both genotypes indicating that ITSN2 inactivation does not compromise B cell survival under these conditions (*Figure 2D*). However, when we examined the expansion of CD95⁺ GL-7⁺ germinal centre (GC) B cells after Vaccinia infection by flow cytometry analysis, we found a 40% reduction in B cells engaged in the GC reaction in *Itsn2⁻/⁻* mice compared with their WT littermates (*Figure 2E*). In contrast, we did not discern a difference in the CD4 T cell compartment of WT versus *Itsn2⁻/⁻* animals; indeed, we observed similar proportions of naïve (CD44⁻CD62L⁺) T cells and activated (CD44⁺CD62L⁻) T cells (*Figure 2F and G*). These experiments show that ITSN2 plays an important role in the generation of B cell responses upon viral infection.

## GC B cells, but not Tfh cells, are impaired in the absence of ITSN2 after vaccination

To further dissect immune responses in an *Itsn2*-deficient context, we immunised WT and *Itsn2⁻/⁻* mice i.p. with the hapten NP conjugated to KLH in alum, a model T cell-dependent antigen, and analysed the splenic B cell compartment 13 days after immunisation. We detected a robust induction of antigen-specific GC B cells (NP⁺CD95⁺GL7⁺) in WT animals, which was 5-fold lower in *Itsn2⁻/⁻* animals (*Figure 3A*). Interestingly, 82% of *Itsn2⁻/⁻* GC B cells harboured a DZ profile in comparison with only 67% of their WT counterparts (*Figure 3B*). In contrast, there was an approximately two-fold reduction in LZ GC fraction in *Itsn2⁻/⁻* compared to WT animals (*Figure 3B*), supporting the notion that ITSN2 is required for B cell progression during the GC reaction. To analyse germinal centre anatomy, we imaged splenic cryosections stained with antibodies against TCRβ (green), B220 (blue) and GL-7 (magenta) to visualise T cell area, B cell follicles and GCs. Interestingly, in sections from *Itsn2⁻/⁻* animals, fewer B cell follicles contained GCs than in the ones from the WT (*Figure 3C*). Moreover, where GCs were present, they were significantly smaller in *Itsn2⁻/⁻* animals than in their WT counterparts (*Figure 3C*).

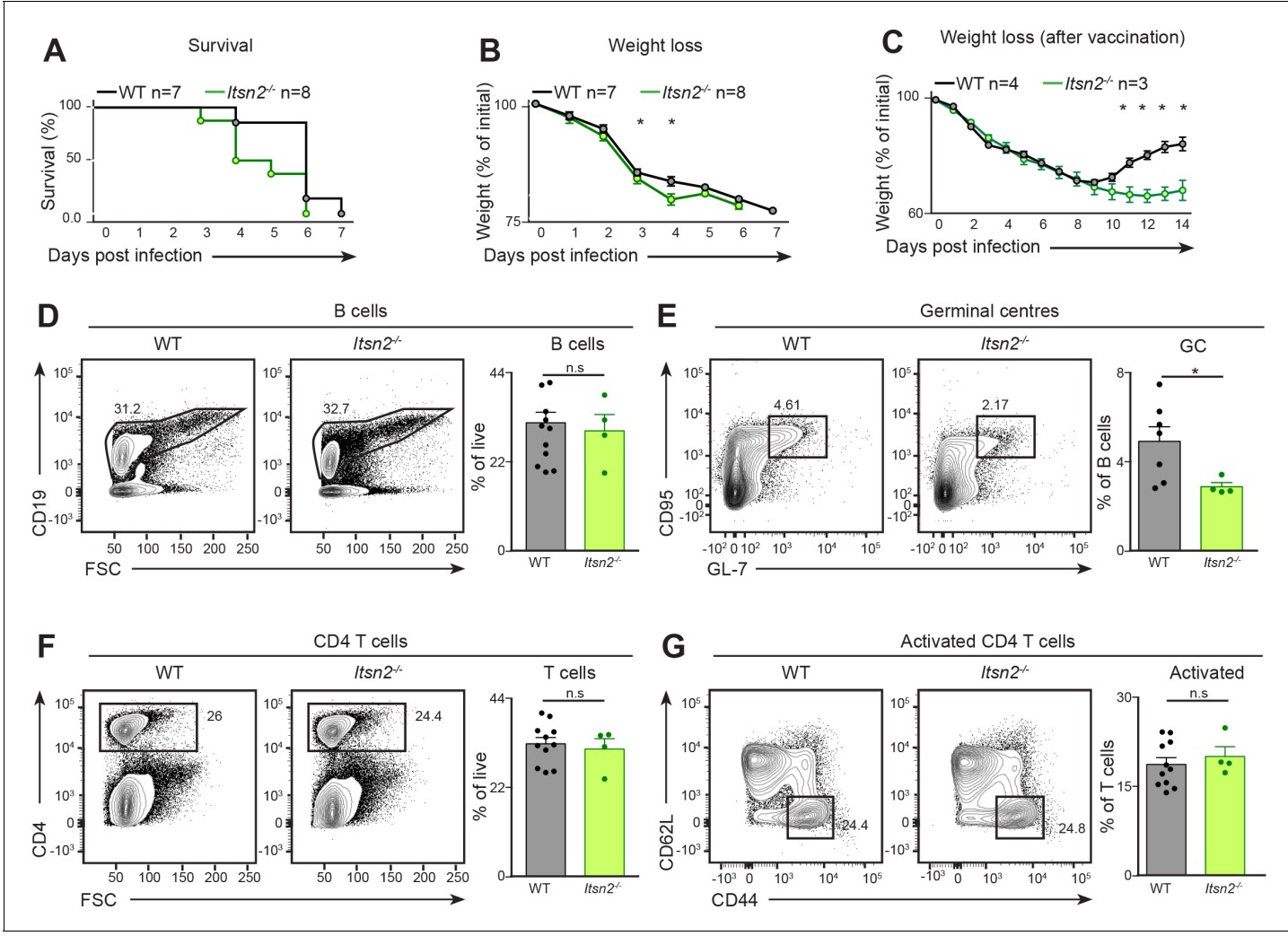

**Figure 2.** *Itsn2⁻ᐟ⁻* animals are impaired in responding to viral infection. (A–B) Age-matched WT and *Itsn2⁻ᐟ⁻* mice were infected intranasally with $3 \times 10^4$ PFU of Influenza virus (PR8 strain). Animals were weighed daily and culled when exhibiting 20% weight loss compared to the weight at Day 0. Survival curve (A) and weight loss (B) as a function of days post infection are shown. Survival analysis, p=0.0345. (C) Age-matched WT and *Itsn2⁻ᐟ⁻* mice were vaccinated by i.v infection of Hemagglutinin trimer with Sigma Adjuvant. After one month, mice were infected with $1.5.10^5$ PFU Influenza PR8 and weighed daily for the whole duration of the experiment. (D–G) WT and *Itsn2⁻ᐟ⁻* littermates were infected with $10^4$ PFU Vaccinia virus (Western Reserve), and draining popliteal lymph nodes were analysed by flow cytometry 7 days post-infection. B cells (CD19⁺, D), GC (B220⁺ GL-7⁺ CD95⁺, E), CD4 T cells (CD4⁺, F) and activated T cells (CD4⁺CD44⁺CD62L⁻, G) are shown. Student's t-test, ns p>0.05, *p<0.05.
DOI: https://doi.org/10.7554/eLife.26556.004

An integral part of a functional GC reaction is the appearance of T follicular helper (Tfh) cells. These are the main subset of T helper (Th) cells that provide costimulatory signals to B cells (*Vinuesa and Cyster, 2011*). In WT animals immunised with NP-KLH, we observed a defined PD1⁺-CXCR5⁺ Tfh population (*Figure 3—figure supplement 1A*). Surprisingly, *Itsn2⁻ᐟ⁻* animals consistently presented a 2-fold increase in the number of Tfh cells compared to WT animals (*Figure 3—figure supplement 1A*). These results indicate that the reduction observed in GC formation upon ITSN2 deletion might not result from lack of Tfh cells, instead it is likely to be associated with intrinsic B cell impairment.

Quantifying the titres of NP-specific antibodies in the serum of immunised mice, we found that titres of the early isotypes IgM and IgG3 were reduced by 2 and 4-fold respectively in *Itsn2⁻ᐟ⁻* mice at d7, but not at d14 (*Figure 3D*). Moreover, at d13 post-immunisation, we detected a 3-fold decrease in the numbers of plasma cells secreting NP-specific IgG in the spleen of *Itsn2⁻ᐟ⁻* mice compared to WT controls, suggesting a potential impairment in the generation of class-switched antibodies

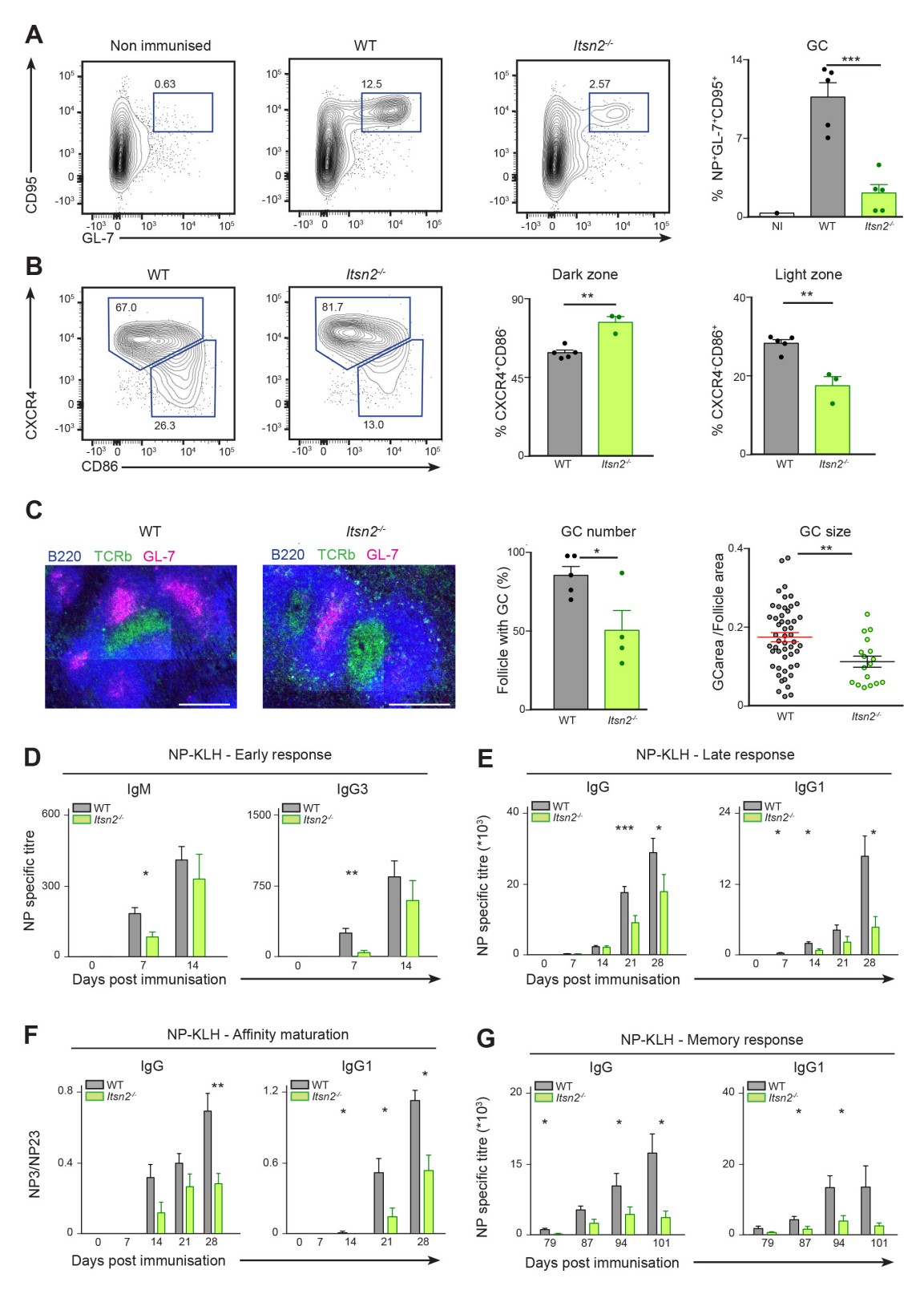

**Figure 3.** Defective humoral responses in *Itsn2*⁻/⁻ animals. WT and *Itsn2*⁻/⁻ littermates were immunised with NP-KLH precipitated in Alum, and B cell responses analysed 13d after immunisation. (**A, B and C**) Spleens of immunised animals were analysed by flow cytometry. Antigen-specific GC B cells (NP⁺ GL-7⁺ CD95⁺, **A**), Dark zone GC B cells (GL-7⁺ CD95⁺ CD86^lo CXCR4^hi, **B**), light zone GC B cells (GL-7⁺ CD95⁺ CD86^hi CXCR4^lo, **B**). Panels on the right show the show percentage of cells in the indicated gates (each dot represents a different animal). (**C**) Tiled images of spleen frozen sections

*Figure 3 continued on next page*

*Figure 3 continued*

acquired by confocal microscopy. GC cells (GL7$^+$, magenta), B cell area (B220$^+$, blue) and T cell area (TCRβ$^+$, green) are shown. Scale bars – 200 μm. Quantification in the two right panels indicate the percentage of B cell follicles with a germinal centre (each dot corresponds to one section), and the ratio between GC area and total follicle area (each dot represents an individual B cell follicle). (D–G) WT and *Itsn2*$^{-/-}$ littermates were immunised with NP-KLH precipitated in Alum. (D) Serum was collected from WT and *Itsn2*$^{-/-}$ mice and titres of NP$_{23}$-specific IgM and IgG3 measured by ELISA. (E) Serum from WT and *Itsn2*$^{-/-}$ mice was collected and titres of NP$_{23}$-specific or total IgG and IgG1 were measured by ELISA. (F) ELISA analysis showing affinity maturation (expressed as the ratio of NP3 to NP23 titres) of total IgG and IgG1 isoform after immunisation. (G) Immunised WT and *Itsn2*$^{-/-}$ animals were submitted to a secondary challenge at day 80 with NP-KLH in PBS, and titres of NP$_{23}$-specific total IgG and IgG1 were measured by ELISA. Data are representative of at least 2 independent experiments with more than three animals in each group. Student's t-test, ns p>0.05, *p<0.05, *p<0.01, ***p<0.0001.

DOI: https://doi.org/10.7554/eLife.26556.005

The following figure supplement is available for figure 3:

**Figure supplement 1.** Defective humoral responses in *Itsn2*$^{-/-}$ animals

DOI: https://doi.org/10.7554/eLife.26556.006

(*Figure 3—figure supplement 1B*). Accordingly, NP-specific IgG and IgG1 titres were significantly reduced (2 and 4-fold respectively) in the serum of *Itsn2*$^{-/-}$ animals compared to WT littermates (*Figure 3E*). To test whether these antibodies had undergone affinity maturation, we compared the binding efficiency of the IgGs present in the sera of immunised mice to NP3-BSA and NP23-BSA. The NP3/NP23 ratio (read-out for affinity maturation) increased steadily in WT mice, but it was two to three time lower in the *Itsn2*$^{-/-}$ mice (*Figure 3F*). This indicates that ITSN2 is required for the generation of high affinity antibodies upon immunisation. To measure the consequences of ITSN2 deletion on B cell memory, animals were submitted to a secondary challenge 3 months after primary immunisation. *Itsn2*$^{-/-}$ animals presented with 2–3-fold decreased memory titres of total IgG and IgG1, and were severely impaired in mounting secondary responses (*Figure 3G*). Taken together, these results show that ITSN2 deletion compromises the production of antigen-specific antibodies in response to T-dependent antigen. To evaluate the function of ITSN2 during T- independent responses, we immunised WT and *Itsn2*$^{-/-}$ mice with NP-LPS, and measured NP-specific antibodies at d4 and d7 post-immunisation. At both time points, titres of IgM, but not IgG3 were significantly reduced in *Itsn2*$^{-/-}$ mice compared to WT controls (*Figure 3—figure supplement 1C*). Together, this set of results highlights the importance of ITSN2 in early and T-independent IgM production, and in the generation of high affinity antibodies upon immunisation.

## Immunodeficiency observed in *Itsn2*$^{-/-}$ animals is associated with B cell intrinsic defects

As the results thus far were obtained with ITSN2 constitutive knockout mice, we wanted to test whether immunodeficiency presented by these animals was associated with B cell intrinsic defects. Accordingly, we generated chimeric mice, wherein lethally irradiated μMT animals lacking mature B cells were injected with a mixture of 80% μMT bone marrow (BM) and 20% WT (WT chimeras) or 20% *Itsn2*$^{-/-}$ (KO chimeras) BM (*Kitamura et al., 1991*). In KO chimeras, all newly generated B cells will be *Itsn2*$^{-/-}$ in an environment containing mainly WT immune cells. After 8 weeks, blood samples were collected to test the reconstitution efficiency by flow cytometry. We observed comparable numbers of B cells in WT and *Itsn2*$^{-/-}$ chimeras, indicating that ITSN2 is not required for bone marrow engraftment under these conditions (*Figure 4—figure supplement 1A*)

*Itsn2*$^{-/-}$ and WT chimeric mice were immunized with NP-KLH in alum and bled weekly for 28 days. We measured antibody titres in the serum of WT and *Itsn2*$^{-/-}$ chimeric mice by ELISA. Noticeably, we observed a severe decrease (3 and 5-fold at d28) in total IgG and IgG1 NP-specific titres in the *Itsn2*$^{-/-}$ compared to the WT chimeric animals (*Figure 4A*). To selectively detect antibodies with high affinity to NP, we used NP4-BSA as capture ligand, and observed a marked decrease in titres of high affinity IgG and IgG1 in *Itsn2*$^{-/-}$ chimeric animals compared to their WT counterparts (*Figure 4B*). These results indicate that selective absence of ITSN2 in the B cell compartment is sufficient to impair antibody responses to T-dependent immunisation.

In the constitutive *Itsn2*$^{-/-}$ mice, impaired antibody production is associated with defective differentiation of GC B cells. To test whether a similar correlation could be established in chimeric mice, we immunised WT and *Itsn2*$^{-/-}$ μMT chimeras with NP-KLH in Alum, and measured B cell responses

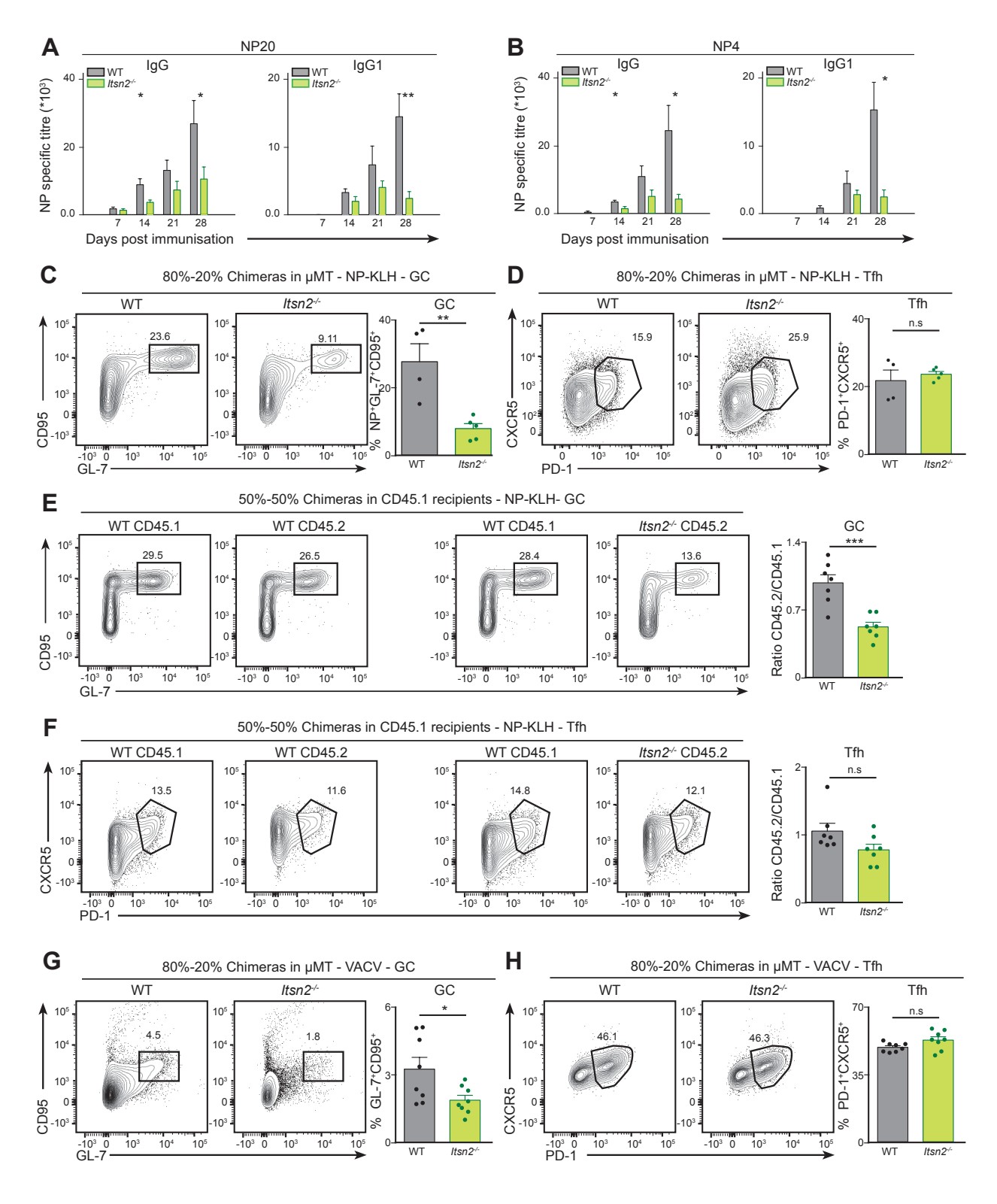

**Figure 4.** *Itsn2⁻/⁻* B cells are intrinsically impaired in responding to immunisation. (**A–C**) Lethally irradiated μMT hosts were reconstituted for 10 weeks with mixtures of 80% μMT BM and 20% WT or *Itsn2⁻/⁻* BM and immunised with NP-KLH in Alum. $NP_{20}$ (**A**) and $NP_4$ (**B**)-specific titres were determined by ELISA. (**C–D**) Spleens of immunised animals were analysed by flow cytometry 13 days after immunisation. Antigen-specific GC B cells ($NP^+$ $GL-7^+$ $CD95^+$, **C**) and Tfh ($CD4^+CXCR5^+PD-1^+$, **D**) are shown. Quantification shown on the right represent the percentage of cells in the indicated gate. (**E–F**)
*Figure 4 continued on next page*

*Figure 4 continued*

Lethally irradiated C57BL6-CD45.1 hosts were reconstituted for 10 weeks with mixtures of 50% WT-CD45.1 BM and 50% WT or *Itsn2*$^{-/-}$ BM, immunised with NP-KLH in Alum, and spleens of immunised animals were analysed by flow cytometry at day 13. CD45.1$^+$ and CD45.2$^+$ NP-specific GC B cells (NP$^+$ GL-7$^+$ CD95$^+$, **E**) and Tfh (CD4$^+$CXCR5$^+$PD-1$^+$, **F**) are shown. Graphs on the right represent the CD45.2/CD45.1 ratio calculated from the percentages of cells in the indicated gates. (**G**) 80–20% mixed chimeras in μMT recipients were infected with 10$^4$ PFU of Vaccinia virus via intra-footpad injections, and popliteal lymph nodes were analysed by flow cytometry at d7. GC (GL-7$^+$ CD95$^+$, **G**) and Tfh cells (GL-7$^+$ CD95$^+$, **H**) are shown. Data are representative of 2–3 independent experiments with at least 4 mice in each group. Student's t-test, ns p>0.05, *p<0.05, *p<0.01, ***p<0.0001.

DOI: https://doi.org/10.7554/eLife.26556.007

The following figure supplements are available for figure 4:

**Figure supplement 1.** Immune compartment reconstitution by *Itsn2*$^{-/-}$ cells

DOI: https://doi.org/10.7554/eLife.26556.008

**Figure supplement 2.** *Itsn2*$^{-/-}$ B cells are intrinsically impaired in responding to immunisation

DOI: https://doi.org/10.7554/eLife.26556.009

in the spleen after 13 days by flow cytometry. In WT chimeras, we detected a robust induction of GC B cells (CD95$^+$GL-7$^+$NP$^+$) whereas in ITSN2 chimeras, the size of this population was 3-fold diminished compared to the WT (*Figure 4C*). However, differentiation of Tfh cells was not impaired in *Itsn2*$^{-/-}$ chimeras compared to WT ones (*Figure 4D*). These results suggest that although *Itsn2*$^{-/-}$ B cells are severely compromised in entering the GC reaction, they are still able to promote the differentiation and survival of Tfh cells as efficiently as WT B cells.

To test how *Itsn2*$^{-/-}$ B cells behaved in competitive settings, we reconstituted lethally irradiated congenic B6.CD45.1 animals with a 50% of CD45.1 BM and 50% of WT CD45.2 (WT-WT) or 50% of *Itsn2*$^{-/-}$ CD45.2 (WT-*Itsn2*$^{-/-}$) BM. The reconstitution efficiency was analysed by flow cytometry 8 weeks after adoptive transfer. While we found comparable proportions of total circulating CD45.2 cells in both types of chimeras, there was a 10% reduction in the fraction of CD45.2 B cells in WT-*Itsn2*$^{-/-}$ versus WT-WT chimeras (*Figure 4—figure supplement 1B and C*). This suggests that under these conditions, WT cells partially outcompete *Itsn2*$^{-/-}$ cells for access to the mature B cell niche. We immunised these animals with NP-KLH in alum and characterised the GC B cell compartment after 13 days. More specifically, we compared the proportion of B cells engaged in the GC reaction in the CD45.1 (WT) and CD45.2 (WT or *Itsn2*$^{-/-}$) in each animal. The CD45.2/CD45.1 ratio was used as a read-out to compare the ability of WT and *Itsn2*$^{-/-}$ B cells to compete with the CD45.1 WT cells present in the same animal. Notably, *Itsn2*$^{-/-}$ B cells were selectively impaired in GC formation (*Figure 4E*), but Tfh differentiation was not affected (*Figure 4F*). These results provide further evidence to support the notion that *Itsn2*$^{-/-}$ B cells are intrinsically impaired in engaging in the germinal centre reaction and in generating high affinity antibodies after immunisation.

To test whether *Itsn2*$^{-/-}$ B cells were also impaired in responding to viral infection, we infected 80–20 chimeras in μMT or 50–50 chimeras in B6.CD45.1 with Vaccinia virus injected intra-footpad and characterised the immune response in the draining popliteal lymph node 7 days later. In both cases, chimeras reconstituted with *Itsn2*$^{-/-}$ bone marrows showed decreased numbers of GC B cells compared to their WT counterparts (*Figure 4G* and *Figure 4—figure supplement 2A*), although the number of Tfh cells was comparable between the two (*Figure 4H* and *Figure 4—figure supplement 2B*).

Together, these results demonstrate that ITSN2 expression in the B cell compartment is required for B cell engagement into the GC response, and for the generation of high affinity antibodies upon immunisation.

## *Itsn2*$^{-/-}$ B cells are defective at BCR signalling

The strong immunodeficiency that we observed in *Itsn2*$^{-/-}$ animals raises the question of whether this molecule is required for biochemical and cellular responses downstream of the BCR. ITSN2 contains several tyrosine residues that can be phosphorylated upon stimulation with estrogen growth factor (EGF) (*Novokhatska et al., 2013*). To check whether BCR ligation caused alterations in ITSN2 phosphorylation status, we stimulated purified naïve B cells with anti-IgM F(ab')$_2$ for 0, 1, 5 or 15 min. Next, cells were lysed and immunoprecipitation was carried out using an anti-phospho-tyrosine antibody. Western blot analysis revealed that both isoforms of ITSN2 were rapidly phosphorylated after

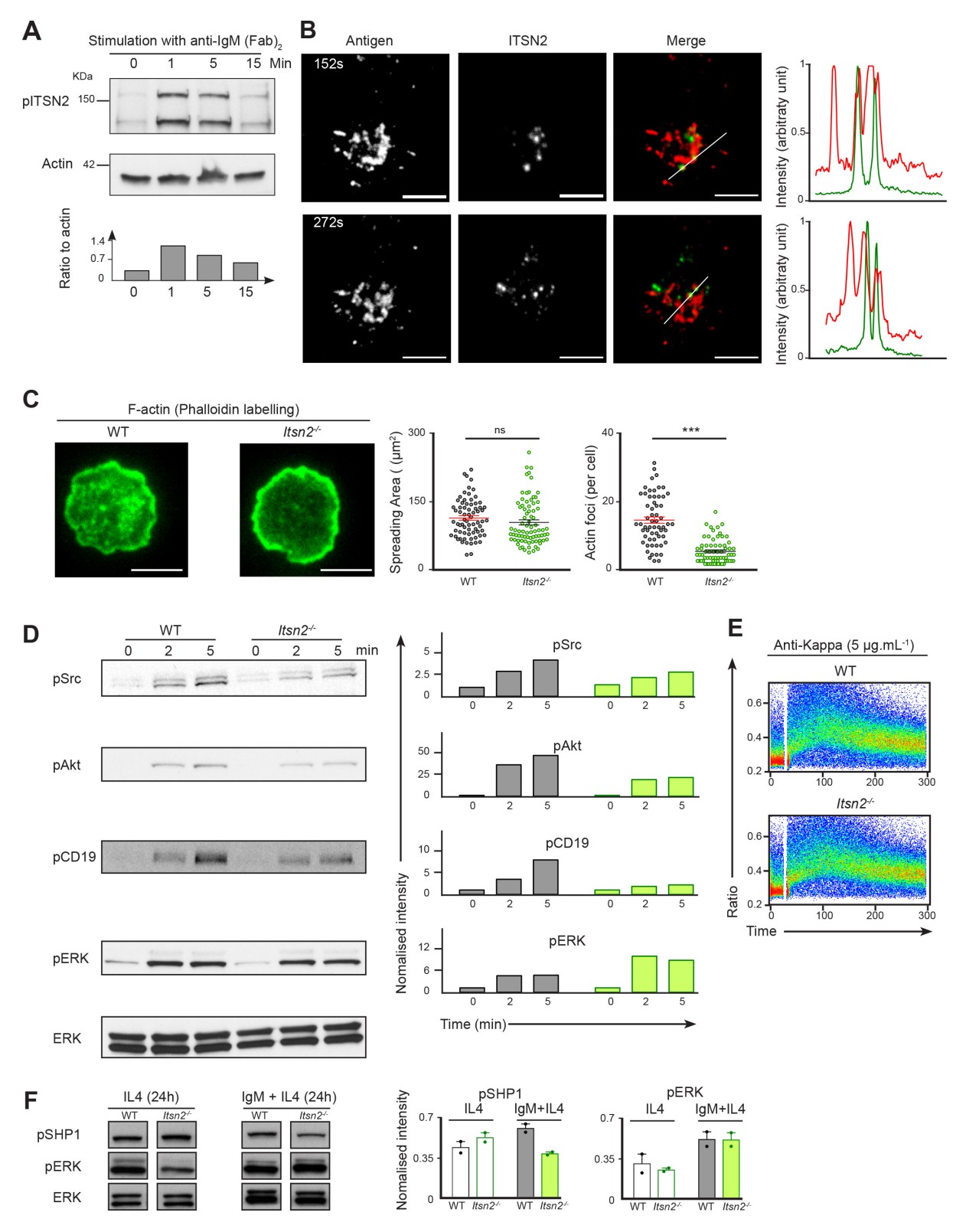

**Figure 5.** ITSN2 participates in the BCR signalling cascade. (**A**) Purified naïve B cells were stimulated with anti-IgM F(ab')₂ (10 µg mL⁻¹) for 0, 1, 5 or 15 min, the total phosphotyrosine fraction was immunoprecipitated, and phosphorylated ITSN2 detected by Western blot. Quantification below shows phosphorylated ITSN2 intensity related to β-Actin. Data are representative of 3 independent experiments. (**B**) A20 cells with a HEL-specific BCR were transfected with a plasmid encoding for ITSN2-L GFP, settled onto fluorescently labelled antigen-loaded lipid bilayers and imaged with a TIRF

*Figure 5 continued on next page*

*Figure 5 continued*

microscope. Localisation of ITSN2 (green) and antigen (red) were compared over time. Scale bars – 3 µm. Data are representative of 3 independent experiments. (C) Representative TIRF images of WT and *Itsn2*[-/-] B cells spread on anti-kappa coated glass coverslips and stained with fluorescently-labelled phalloidin. Scale bars – 3 µm. The quantification of spreading area and number of detected actin foci are shown graphically. Each dot represents an individual cell. Data are representative of 3 independent experiments where > 100 cells of each genotype were analysed. (D) WT and *Itsn2*[-/-] B cells were stimulated with 2 µg mL$^{-1}$ anti-IgM F(ab')2 for 0, 2 or 5 min, and phosphorylation of Src, CD19, Akt (S473) and ERK were detected by Western blotting. Graphs next to each lane indicate the relative intensity of each band relative to ERK at t0. (E) Intracellular calcium influx in purified WT or *Itsn2*[-/-] B cells after stimulation with 5 µg mL$^{-1}$ anti-kappa. (F) Naïve purified WT and *Itsn2*[-/-] B cells were cultured in the presence of IL-4 alone or IL-4 and anti-IgM for 24 hr, and phosphorylation ERK and SHP1 measured by Western Blot. Student's t-test, ns p>0.05, *p<0.05, **p<0.01, ***p<0.001.
DOI: https://doi.org/10.7554/eLife.26556.010

The following figure supplements are available for figure 5:

**Figure supplement 1.** ITSN2 participates in the BCR signalling cascade.
DOI: https://doi.org/10.7554/eLife.26556.011

**Figure supplement 2.** B cell activation ex vivo in response to BCR or TLR signals is not impaired in *Itsn2*[-/-] B cells.
DOI: https://doi.org/10.7554/eLife.26556.012

BCR stimulation (*Figure 5A*), suggesting a potential role for the phosphorylation of this molecule in the BCR signalling cascade.

To test whether ITSN2 was recruited to the active sites of BCR signalling, we transfected A20 cells carrying a HEL-specific transgenic BCR (with the specificity of D1.3) with a construct expressing ITSN2L-GFP. These cells were allowed to settle onto planar lipid bilayers loaded with Alexa Fluor 633-labeled HEL, mimicking B cell interaction with APCs, and imaged by TIRF microscopy. Interestingly, as the cells spread over the bilayers and formed antigen microclusters, we observed the appearance of distinct ITSN2 foci in close proximity to antigen (*Figure 5B*). However, we did not detect a direct colocalisation between ITSN2 and antigen microclusters. These results indicate that the role exerted by ITSN2 in immunity is independent of its physical recruitment to antigen microclusters, and instead may be through its ability to signal downstream effectors.

Given the early ITSN2 recruitment, we wanted to establish whether its absence influences the re-organisation of the actin cytoskeleton after BCR activation. WT and *Itsn2*[-/-] B cells were allowed to spread on coverslips coated with immobilised antigen and stained for F-actin with phalloidin. WT B cells spread over the antigen-coated surface and exhibited a typical F-actin distribution, with a characteristic ring at the periphery of the cells, and about 14 actin foci per WT cell (*Figure 5C*). Although the overall spreading area was not affected in *Itsn2*[-/-] B cells, the organisation of the F-actin network was altered (*Figure 5C*) as evident by a dramatic reduction in these foci with an average of 4 F-actin foci per cell. This is reminiscent of what has been previously reported for other components of the Wiskott-Aldrich syndrome protein complex (*Keppler et al., 2015*; *Kumari et al., 2015*) (*Figure 5C*). We also observed a reduction in actin foci in two independent CRISPR/Cas9 inactivated *Itsn2*[-/-] A20 lymphoma B cell lines expressing a HEL-specific BCR (*Ran et al., 2013*). In these cell lines, actin foci formation was restored by overexpression of ITSN2LGFP, indicating that this defect is indeed due to ITSN2 deficiency (*Figure 5—figure supplement 1A*).

To determine whether this defect in actin cytoskeleton might reflect impaired BCR signalling, we stimulated WT and *Itsn2*[-/-] B cells anti-IgM F(ab')2 for 0, 2 or 5 min and measured the phosphorylation status of several canonical effectors by Western blotting. Both WT and *Itsn2*[-/-] B cells exhibited a robust phosphorylation of the mitogen-activated protein kinase (MAPK) ERK upon BCR engagement (*Figure 5D*). However, phosphorylation of Src, Akt and CD19 was reduced by at least 2-fold in *Itsn2*[-/-] B cells compared to WT (*Figure 5D*). This suggests that ITSN2 is implicated in propagating BCR-driven signals, in particular those leading to activation of the PI3K pathway. However, these early defects did not impair calcium influx, as both WT and *Itsn2*[-/-] B cells exhibited robust calcium flux upon BCR engagement (*Figure 5E*).

We recently described that WIP-deficient B cells exhibit strong defects in activation of the PI3K pathway, in particular in CD19 phosphorylation, coupled to impaired actin foci formation (*Keppler et al., 2015*). These defects are coupled to an increase in BCR and CD19 motility at the single molecule level. Measuring the diffusion coefficients of IgM, IgD and CD19 in *Itsn2*[-/-] B cells, we observed a small increase in CD19 mobility, but much less marked than in the case of the WIP KO (*Figure 5—figure supplement 1B*).

To evaluate the consequence of impaired early BCR signalling in *Itsn2^-/-* B cells on distal signalling events, we cultured WT and *Itsn2^-/-* B cells in the presence of IL-4 with or without anti-IgM F(ab')2. After 24 hr, cells were recovered and the phosphorylation of effector proteins was detected by Western blotting. Notably, in the condition where anti-IgM was added, we observed a consistent 40% reduction in levels of phosphorylated SHP1 in *Itsn2^-/-* B cells compared to their WT counterparts (**Figure 5F**). In contrast pERK levels were not affected by ITSN2 deletion. This shows that indeed, ITSN2 deficiency leads to sustained signalling defects upon BCR engagement.

To gain insight into the physiological consequences of impaired signalling on B cell survival and activation, we measured the survival of cultured naïve WT and *Itsn2^-/-* B cells after BCR cross-linking. The results of this experiment revealed that cell survival was comparable in both cell types (**Figure 5—figure supplement 2A**, left panel). Moreover, when IL-4 and IL-5 were added to the culture medium, both WT and *Itsn2^-/-* B cells underwent robust proliferation as highlighted by the multiple peaks of CTV dilution (**Figure 5—figure supplement 2B**), indicating that in spite of their signalling impairment, *Itsn2^-/-* B cells can be activated ex vivo. Next we compared the activation of WT and *Itsn2^-/-* B cells in response to stimulation through TLR4, TLR9 or CD40, and found that in all tested conditions, both cell types proliferated to the same extent (**Figure 5—figure supplement 2C and D**). To evaluate the role of ITSN2 in integrating BCR and TLR9 signals, we stimulated naïve WT and *Itsn2^-/-* B cells with particles coated with IgM and CpG, and measured proliferation after 3 days. We observed comparable proliferation patterns for WT and *Itsn2^-/-* B cells (**Figure 5—figure supplement 2E**). These results indicate that ITSN2 deletion does not compromise B cell proliferation ex vivo in response to a range of soluble stimuli. Together, these results indicate that although ITSN2 deletion alters the actin cytoskeleton and PI3K signalling upon BCR stimulation, it does not seem to compromise B cell survival and proliferation in response to a range of soluble stimuli ex vivo.

## *Itsn2^-/-* B cells are impaired in harnessing T cell help in vivo

Antigen binding to the BCR triggers its internalisation, processing and presentation on MHCII (**Amigorena et al., 1994**). This allows antigen-primed B cells to interact with cognate CD4 T cells that provide them co-stimulatory signals required for their full activation (**Lanzavecchia, 1985**). We first carried out a series of ex vivo assays in which we tested the effectiveness of *Itsn2^-/-* B cells in harnessing T cell help. We first measured internalisation of soluble and particulate antigen, and found similar rates for WT and *Itsn2^-/-* B cells (**Figure 6—figure supplement 1A and B**). Next, we stimulated cells with particles coated with anti-IgM and Eα peptide to follow antigen presentation (**Rudensky AYu et al., 1991**), and detected comparable levels of MHCII-Eα conjugates on both cell types after 5 hr (**Figure 6—figure supplement 1C**). Finally, we stimulated WT and *Itsn2^-/-* B cells with microspheres coated with anti-IgM antibodies and OVA, and co-cultured them with OTII T cells (**Barnden et al., 1998**). We detected similar expression of activation markers (CD40, CD69 and CD86) by cells from both genotypes 24 and 48 hr after stimulation, and B and T cells proliferated with comparable rates (**Figure 6—figure supplement 1D and E**). These results indicate that ITSN2 is not required for BCR-driven antigen internalisation or for antigen presentation, and that *Itsn2^-/-* B cells can activate cognate T cells and sense costimulation ex vivo.

These ex vivo results do not reflect the significant germinal centre and antibody phenotype observed in *Itsn2^-/-* animals in vivo. There are some parallels to this, for example in the case of CD84 deficiency, that severely impairs humoral responses in vivo, but not T cell proliferation and activation ex vivo (**Cannons et al., 2010**). Animals lacking the SLAM associated protein (SAP) also present with drastic in vivo defects that do not correlate with marked defects in T cell activation in vitro (**Cannons et al., 2006**; **Qi et al., 2008**). Therefore, we sought to analyse some aspects of the activation of *Itsn2^-/-* B cells in vivo. To this end, we crossed *Itsn2^-/-* mice to MD4-Tg expressing a BCR with high affinity to HEL (**Goodnow et al., 1988**), thereafter referred to as HEL-*Itsn2^-/-*). We next purified HEL-WT and HEL-*Itsn2^-/-* B cells, labelled them with CTV, and transferred them to C57BL/6 CD45.1 recipients together with CFSE-labelled OTII T cells. Following adoptive transfer, we immunised mice hosts with HEL and OVA coated microspheres and analysed their spleens by flow cytometry 3 days later. Under these conditions, WT B cells underwent intense proliferation, as highlighted by the multiple peaks of CTV dilution. Strikingly, *Itsn2^-/-* B cells proliferated to a lesser extent than WT cells (**Figure 6A**), indicating that ITSN2 deletion compromises B cell activation in vivo.

To test whether the impaired activation of *Itsn2^-/-* B cells derived from defects in antigen uptake or presentation in vivo, we transferred CTV-labelled HEL-WT and CFSE-labelled HEL-*Itsn2^-/-* B cells

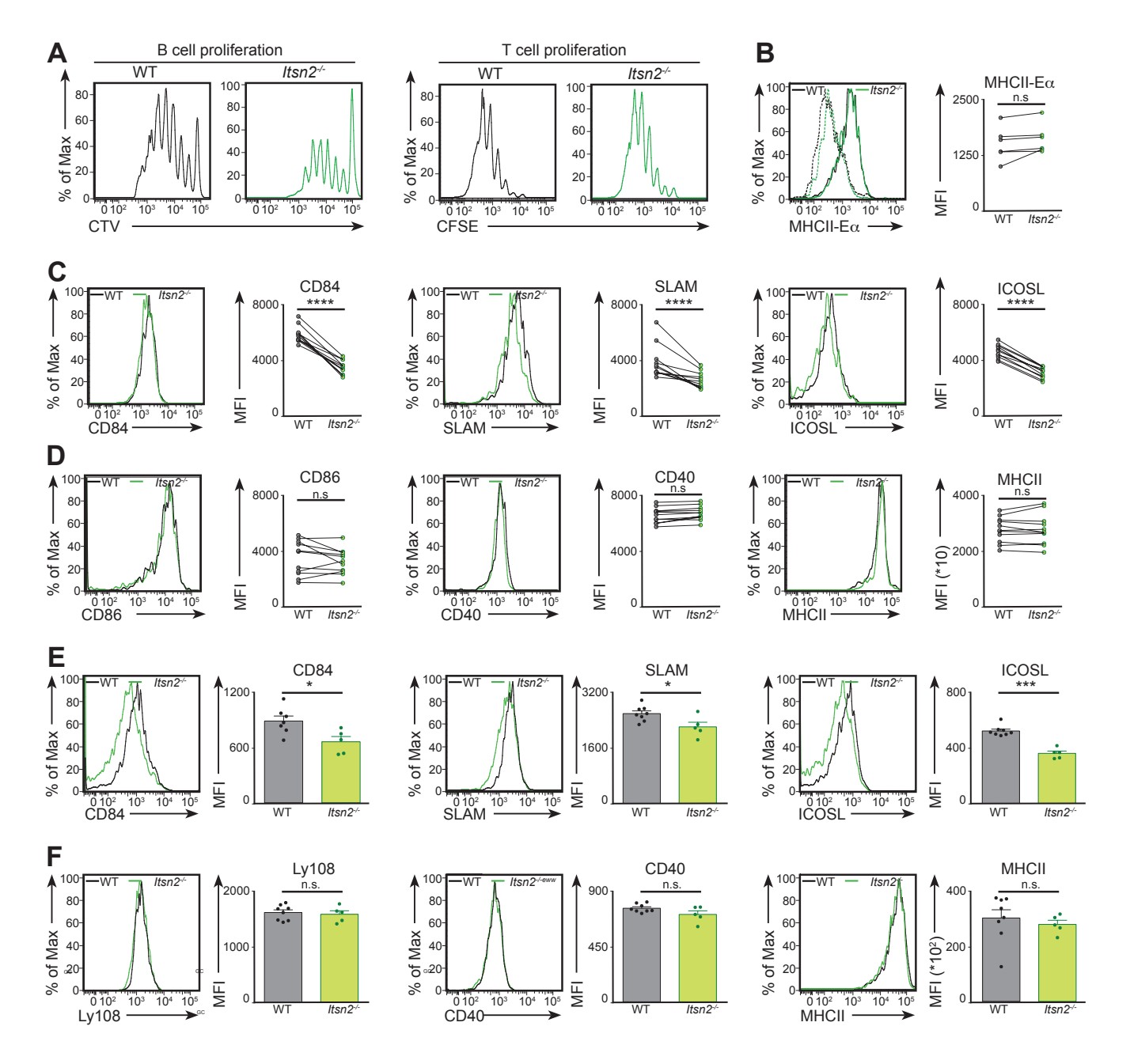

**Figure 6.** ITSN2 is required for B cell activation in vivo. (**A**) CTV-labelled HEL-WT or HEL-*Itsn2⁻/⁻* B cells and CFSE-labelled OTII T cells were transferred to CD45.1 hosts, subsequently immunised with HEL and OVA coated microspheres. Proliferation of both cell types was measured by flow cytometry 3 days after immunisation. Data are representative of 4 independent experiments with 3–6 animals in each group. (**B**) CTV-labelled HEL-WT and CFSE-labelled HEL-*Itsn2⁻/⁻* B cells were transferred to WT recipients prior to intravenous immunisation with beads coated with HEL and Eα peptide. Expression of MHCII-Eαconjugates (quantified on the right as the geometric mean of the MHCII- Eα fluorescence signal) by both cell types was measured 24 hr after immunisation. Data are representative of 2 independent experiments with 5–10 mice. (**C–F**) CFSE-labelled HEL-WT and CTV-labelled HEL-*Itsn2⁻/⁻* B cells were transferred to CD45.1 hosts, and immunised with HEL and OVA coated microspheres. Expression of CD84 (**C**), SLAM (**C**), ICOSL (**C**), CD86 (**D**), CD40 (**D**) and MHCII (**D**) was measured by flow cytometry 24 hr after immunisation. Data are representative of 3 independent experiments with 5–10 mice. (**E–F**) WT and *Itsn2⁻/⁻* littermates were immunised with NP-KLH precipitated in Alum, and spleens analysed by flow cytometry 13 days after immunisation. Expression of CD84 (**E**), SLAM (**E**), ICOSL (**E**), Ly108 (**F**), CD40 (**F**) and MHCII (**F**) on germinal centre B cells was measured. Data are representative of 3 independent experiments with 5–10 mice in each group. Paired t-test (**B, C, D**) or Student's t-test (**E,F**), ns $p > 0.05$, *$p < 0.05$, ***$p < 0.001$, ****$p < 0.0001$.

DOI: https://doi.org/10.7554/eLife.26556.013

*Figure 6 continued on next page*

*Figure 6 continued*

The following figure supplement is available for figure 6:

**Figure supplement 1.** Antigen uptake and reception of T cell help ex vivois not impaired in *Itsn2-/-* B cells.
DOI: https://doi.org/10.7554/eLife.26556.014

to WT recipients prior to immunisation with fluorescent beads coated with HEL and Eα. We found similar numbers of WT and *Itsn2-/-* B cells positive for beads, suggesting that ITSN2 is not required for antigen uptake in vivo (*Figure 6—figure supplement 1F*). Next, we measured presentation of Eα peptide on MHCII by flow cytometry 20 hr after antigenic challenge in vivo. In these conditions, beads-positive WT and *Itsn2-/-* B cells expressed comparable amounts of MHCII- Eα conjugates (*Figure 6B*), indicating that ITSN2 is not required for antigen capture and presentation in vivo.

We next measured the expression of various surface receptors on activated B cells 20 hr after immunisation. Strikingly, we found that the expression of the immune receptors CD84 and SLAM, and of the inducible co-stimulator ligand (ICOSL) were significantly decreased in *Itsn2-/-* B cells compared to WT levels (*Figure 6C*). Both cell types expressed similar levels of the co-stimulatory molecules CD40 and CD86, as well as MHCII (*Figure 6D*). This shows that ITSN2 is required for the upregulation of several surface receptors after antigen challenge in vivo.

To determine whether similar discrepancies could be observed at later stages during the immune response, we immunised WT and *Itsn2-/-* animals with NP-KLH in alum and measured the expression of various markers by GC B cells after 13 days. Interestingly, *Itsn2-/-* GC B cells expressed reduced levels of CD84, SLAM, and ICOS-L compared to their WT counterparts (*Figure 6E*). In contrast, expression of the costimulatory receptor CD40, the negative regulator Ly108 and MHCII were comparable in WT and *Itsn2-/-* B cells (*Figure 6F*).

This set of experimental results suggests deletion of ITSN2 leads to alterations in the expression of various receptors upregulated upon B cell activation in vivo, which were maintained when the cells progressed into the GC reaction. A common feature of the receptors whose expression is deregulated in *Itsn2-/-* B cells is their involvement in mediating interactions between B cells and cognate T cells (*Crotty, 2015*). ICOSL binds to ICOS expressed on CD4 T cells, and participates in instructing T cells to differentiate into Tfh cells (*Liu et al., 2015*); while CD84 and SLAM are expressed by both B and T cells and can engage into homotypic interactions, thus participating to the establishment of B-T synapses (*Schwartzberg et al., 2009*). Notably, SAP-deficient T cells also present with altered expression of surface receptors, with markedly reduced ICOS expression upon stimulation (*Cannons et al., 2006*). This correlates with shortened contacts with B cells in vivo (*Qi et al., 2008*). These observations prompted us to further characterise the function of ITSN2, in the context of cognate B-T interactions in vivo.

## ITSN2 is required for the establishment of long-term interactions between B and T cells

To study the impact of ITSN2 deficiency on B cell behaviour in vivo, CFSE-labelled WT and CTV-labelled *Itsn2-/-* B cells were transferred to WT recipients. After 24 hr, popliteal and inguinal lymph nodes were explanted and imaged by multiphoton microscopy. We observed that both cell types were found in B cell follicles in comparable proportions, they moved at an average speed of 5 μm min$^{-1}$, and exhibited the typical random-walk behaviour described for naïve B cells (*Miller et al., 2002*) (*Figure 7A and B*, and *Figure 7—video 1*). This indicates that *Itsn2-/-* B cells are able to home to skin draining lymph nodes as efficiently as WT B cells and that ITSN2 is not required for B cell motility in steady-state conditions.

Upon binding to cognate antigen, B cells adopt a different behaviour, and migrate towards the B-T border where encounter with CD4 T cells is more likely. To test whether ITSN2 was involved for B cell relocalisation upon activation, we transferred CTV-labelled HEL-WT, and CFSE-labelled HEL-*Itsn2-/-* to WT recipients. After 24 hr, some animals were immunised with microspheres coated with HEL and OVA via intra-footpad injections, while others received PBS injections as a control. We explanted popliteal lymph nodes 20 hr after immunisation and used them to prepare cryosections, which were stained with antibodies against B220 and TCRβand imaged with a confocal microscope. Using this method, we were able to analyse the localisation of WT and *Itsn2-/-*B cells within the LN

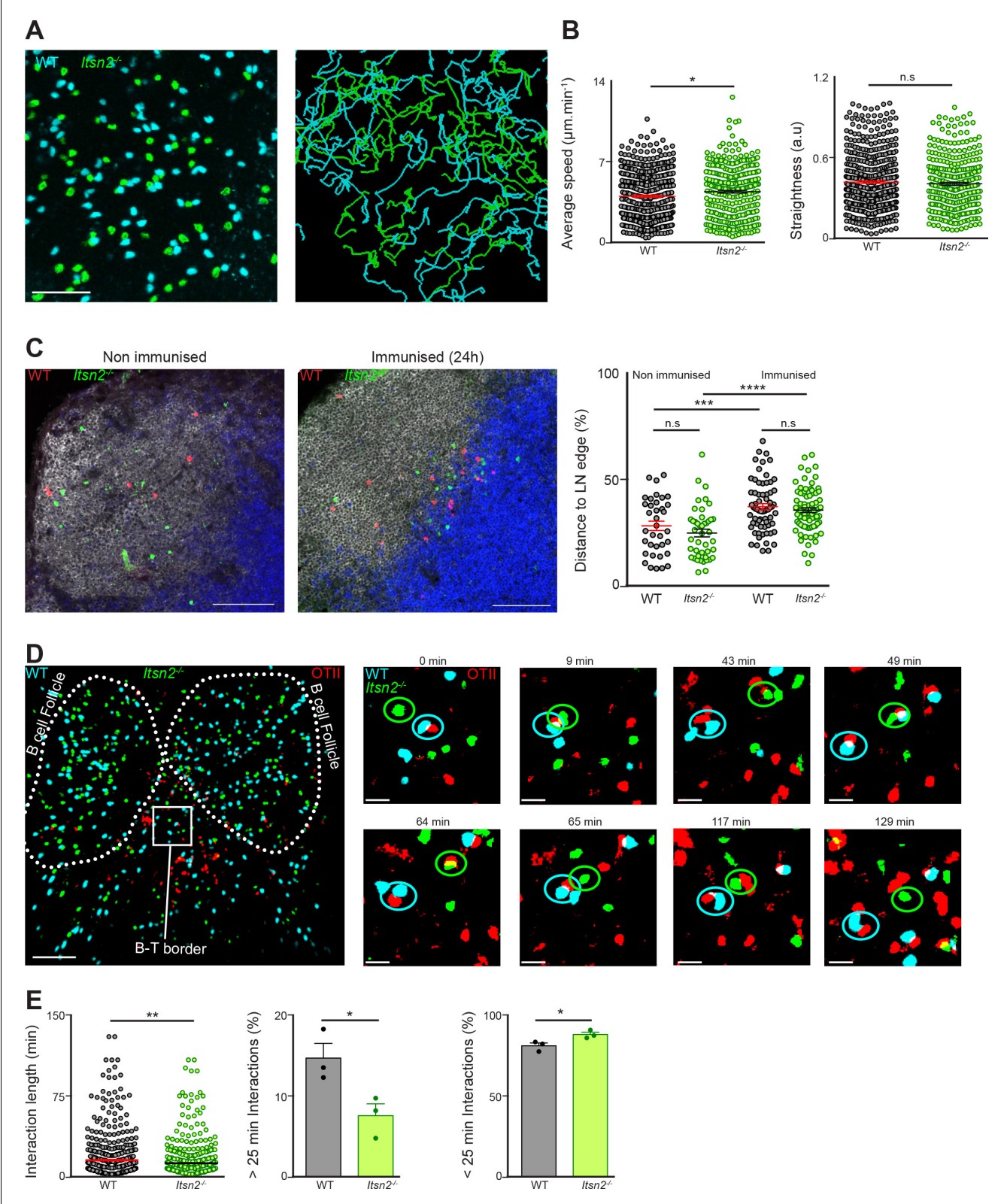

**Figure 7.** *Itsn2⁻/⁻* B cells are impaired in forming long-term conjugates with T cells. (**A–B**) CTV-labelled HEL-WT and CFSE-labelled-*Itsn2⁻/⁻* B cells were transferred to WT recipients, and explanted popliteal lymph nodes were imaged with a multiphoton microscope 24 hr after adoptive transfer. A representative field of view as well as the associated tracks are shown. (**B**) Quantification representing the average speed and the straightness of the tracks. Data are representative of 2 independent experiments where 4–6 lymph nodes were imaged. Student-s t-test, n.s p>0.05, *p<0.05. (**C–E**) CTV-
*Figure 7 continued on next page*

*Figure 7 continued*

labelled HEL-WT and CFSE-labelled HEL-*Itsn2⁻/⁻* B cells were transferred to WT recipients together with unlabelled (C) or SNARF-1-labelled (D–E) OTII T cells prior to intra-footpad immunisation with HEL and OVA coated microspheres. Popliteal lymph nodes were explanted 18–20 hr later imaged by confocal (C) or multiphoton microscopy (D–E). (C) Frozen sections were stained with antibodies against B220 and TCRβ to distinguish B and T cell areas. Quantified data on the right indicate the distance from each WT (red) or *Itsn2⁻/⁻* (green) B cell to the edge of the LN. Scale bars – 200 µm. Paired t-test, n.s. – p>0.05). (D) Representative images of a whole popliteal lymph node as well as higher magnification images showing typical interactions between WT B cells (cyan), *Itsn2⁻/⁻* B cells (green) and OTII T cells (red). Dotted lines were added for illustrative purposes and indicate the localisation of two B cell follicles. (Scale bar – 80 µm on the low magnification and 10 µm on the higher magnification.) (E) Quantification of the overall length of B-T interactions (left panel) and the fraction of interactions shorter (middle panel) or longer (right panel) than 25 min. Data were pooled from 3 independent experiments where 2–3 lymph nodes were imaged. Student's t-test, ns p>0.05, *p<0.05, **p<0.01.

DOI: https://doi.org/10.7554/eLife.26556.015

The following videos are available for figure 7:

**Figure 7—video 1.** Representative movie of part of a popliteal lymph node, showing homeostatic migration of WT (gray) and *Itsn2⁻/⁻* (green) B cells.

DOI: https://doi.org/10.7554/eLife.26556.016

**Figure 7—video 2.** Representative movie of part of a popliteal lymph node, showing long term interaction between a WT B cell (cyan) and OTII T cells (red), and shorter conjugates between *Itsn2⁻/⁻* B cells (green) and OTII T cells (red).

DOI: https://doi.org/10.7554/eLife.26556.017

before and after immunisation. In naïve recipients, both cell types were found predominantly within B cell follicles (*Figure 7C*). As expected, immunisation with cognate antigen triggered the displacement of B cells towards the B-T border with *Itsn2⁻/⁻* B cells phenocopying the WT, indicating that ITSN2 is not required for B cell relocalisation upon activation (*Figure 7C*).

To characterise cognate interactions between activated T and B cells in vivo, we transferred labelled HEL-WT, HEL-*Itsn2⁻/⁻* and OTII T cells to WT recipients prior to intra-footpad immunisation with microspheres coated with HEL and OVA. 18 hr after immunisation, we explanted the draining popliteal lymph nodes and imaged them with a multiphoton microscope to visualise the formation of B–T conjugates. WT B cells formed conjugates with OTII T cells, many of which were stable and lasted for longer than 25 min (*Figure 7D and E* and *Figure 7—video 2*). In contrast, although *Itsn2⁻/⁻* B cells did form conjugates with OTII T cells, the average length of interaction was reduced compared to those of WT B cells. Moreover, we observed a significant decrease in the fraction of *Itsn2⁻/⁻* B cells that interact with T cells for longer than 25 min (*Figure 7D and E* and *Figure 7—video 2*). These results show that ITSN2 expression in B cells is essential for the formation of long-lived cognate B–T conjugates.

Together, this study highlights the role of ITSN2 in the differentiation of GC B cells and the production of high affinity antibodies upon immunisation. The molecular mechanism for this resides in the fact that *Itsn2⁻/⁻* B cells fail to upregulate several receptors upon activation, which severely compromises their ability to form stable conjugates with cognate T cells, and hence their full activation and entry into germinal centres.

## Discussion

In this study, through a combination of immunisation and infection experiments in mice lacking ITSN2, we have provided the first analysis of the role of this adaptor protein during immune responses. ITSN2 deficiency hindered early signalling downstream of BCR engagement and cytoskeleton remodelling triggered by antigen recognition, but did not compromise B cell activation ex vivo. However, ITSN2 deletion in B cells in vivo resulted in mice with severely impaired formation of GC B cells, reduced antibody responses and incomplete protection after Flu vaccination. Furthermore, in vivo, activation of *Itsn2⁻/⁻* B cells was impaired, and the cells expressed reduced amounts of surface receptors involved in mediating interactions between B and T cells. As a consequence, ITSN2 deficiency in B cells impaired the establishment of long term conjugates with cognate T cells.

The characterisation of the adaptive immune response during viral infection or immunisation highlighted that ITSN2 deletion causes severe defects in the differentiation of GC B cells and production of high affinity antibodies. These results are in line with the role of ITSN2-interaction partners WIP, WASp, and N-WASP in B cell function; indeed, B cells lacking WIP or both WASp and N-WASp are also impaired in entering the GC reaction and producing antibodies after vaccination (*Keppler et al., 2015*; *Westerberg et al., 2012*). Similarly, ablation of DOCK8, another

multimodular adaptor with GEF activity, in the B cell compartment causes defects in the generation of long term antibody responses (*Randall et al., 2009*; *Ruusala and Aspenström, 2004*). Notably, mutations in *WAS*, WIPF1 or DOCK8 have been reported in human immunodeficiencies (*Sullivan et al., 1994*; *Zhang et al., 2009*), underscoring the importance of these cytoskeleton regulators during immune responses. Interestingly, ITSN2 has been identified as a potential candidate for primary immunodeficiency in a bioinformatics screen (*Keerthikumar et al., 2009*). Moreover, ITSN2 was found to be overexpressed in B lymphocytes from patients with Sjögren's syndrome (*Lessard et al., 2013*; *Miceli-Richard et al., 2016*).

The BCR engagement of membrane-tethered ligands leads to the formation of antigen microclusters and triggers a complex signalling cascade (*Kurosaki et al., 2010*). Here, we found that ITSN2 is recruited to the plasma membrane upon BCR engagement, and although it is not directly recruited to antigen microclusters, both ITSN2 isoforms were rapidly phosphorylated on tyrosine residues after antigen binding to the BCR. In various cell lines, ITSN2 phosphorylation has been reported in response to stimulation by estrogen growth factor (*Novokhatska et al., 2013*). Antigen engagement by the BCR leads to the activation of the Src family kinases and Syk, leading to tyrosine phosphorylation of multiple effectors, including Igα and Igβ (*Kurosaki et al., 2010*). This promotes the recruitment of proteins containing Src homology (SH2) domains that play a central role in propagating BCR signalling (*Dal Porto et al., 2004*). Interestingly, ITSN2 was found to interact with the SH2 domains of multiple proteins, such as Fyn, Grb2 or else PLCγ2 (*Novokhatska et al., 2013*). However, the precise functional consequences of ITSN2 phosphorylation in the context of BCR stimulation are still to be investigated in detail.

*Itsn2*$^{-/-}$ B cells were selectively impaired in forming actin foci in response to BCR signalling. Interestingly, WIP deficiency in B cells also compromises the appearance of these actin nodes (*Keppler et al., 2015*). In T cells, similar actin-rich structures have been proposed to be sites of WASp-dependent actin polymerisation, to participate in PLCγ1 activation, and were missing in T cells deficient in WASp and N-WASp (*Kumari et al., 2015*). Thus, the similarity of the defects in actin morphology caused by deletion of ITSN2, WIP and WASp interrogates whether these proteins participate in the same signalling axis to regulate actin remodelling after antigen engagement of the BCR. Such interconnection has been reported in the context of Vaccinia virus infection, during which ITSN1, the other member of the intersectin family, was reported to collaborate with Nck and Cdc42 to promote N-WASp dependent actin polymerisation (*Humphries et al., 2014*).

Antigen binding to the BCR leads to antigen internalization into endosomal compartments, trafficking of the internalized antigen to lysosomes for proteolytic cleavage, loading of peptide antigens onto MHCII molecules, and subsequent presentation to cognate T cells (*Amigorena et al., 1994*; *Lanzavecchia, 1985*). We found that B cells lacking ITSN2 were as effective as WT cells in antigen internalization and presentation, indicating that ITSN2 is dispensable for these processes. In the context of T-dependent responses, antigen presentation by B cells is critical for the establishment of the Tfh population. Tfh cells are strictly required for the generation of high-affinity, class-switched antibodies and for the establishment of B cell memory (*Vinuesa and Cyster, 2011*). Tfh differentiation and function relies on multiple signals, including contact-dependent interactions between TCR and MHCII-peptide conjugates, ICOS or CD28 and their respective ligands ICOS-L and CD80/CD86, and soluble signals derived from cytokines, such as IL21 or IL6 (*Crotty, 2015*). Notably, animals deficient for ITSN2 show reduced numbers of B cells engaged in the GC reaction, but comparable or increased numbers of Tfh cells. This may be due to the fact that antigen presentation is not defective in *Itsn2*$^{-/-}$ B cells.

Interactions between GC B cells and Tfh cells occur predominantly in the GC light zone, and are key for the affinity-based selection of GC B cells. Hypermutated GC B cells with higher affinity capture more antigen from FDCs than lower affinity variants, present more MHCII-peptide conjugates, and form longer interactions with cognate Tfh cells (*Shulman et al., 2013*). As a result, they receive stronger signals and have a proliferative advantage after re-entering the dark zone (*Gitlin et al., 2014*). In *Itsn2*$^{-/-}$ mice, the GC cells were biased to the dark zone, with fewer cells exhibiting the light zone markers. This may be explained by defects *Itsn2*$^{-/-}$ B cells presented in establishing long-term interactions with cognate T cells.

ITSN2 deficiency in B cells was associated with reduced length of cognate B-T interactions. In mouse, SAP deletion dramatically impairs antibody responses, as SAP KO T cells fail to establish long term contacts with cognate B cells, and are not retained in the GC (*Crotty et al., 2003*;

*Qi et al., 2008*). SAP is a small adaptor protein mediating signals downstream of receptors of the SLAM family, such as CD84, SLAM or Ly108. T cell intrinsic deficiency in CD84, but not in SLAM or Ly108, is associated to reduced GC formation and to shortened interactions between B and T cells, but without recapitulating the phenotype of SAP inactivation (*Cannons et al., 2010*; *2006*; *Kageyama et al., 2012*; *McCausland et al., 2007*).

The deletion of ITSN2 severely impaired B cell activation in vivo and was associated to decreased expression of various surface receptors both shortly after immunisation and on GC B cells. These results intriguingly mirrors the correlation observed in SAP-deficient T cells between decreased ICOS expression, and impaired formation of B-T conjugates (*Cannons et al., 2006*; *Qi et al., 2008*). In particular, *Itsn2⁻/⁻* B cells displayed reduced levels of SLAM and CD84, two receptors of the SLAM family, and of ICOSL. This is particularly relevant, since IL4 production by Tfh cells is SLAM dependent (*Yusuf et al., 2010*), while that of IL21 is dependent on ICOS triggering (*Morita et al., 2011*). The reduced expression of these molecules by in ITSN2 GC B cells could result in reduced cytokine secretion by Tfh cells, and hence hinder their reception of T cell help.

In contrast, expression of Ly108, another member of the SLAM family, was unaffected by ITSN2 deletion. CD84, SLAM and Ly108 are upregulated by B cells upon activation (*Cannons et al., 2010*). They can all signal through the intracellular adaptor SAP, and function by engaging into homotypic interactions (*Latour et al., 2001*). Interestingly, the singularity of Ly108 function is its ability to mediate a strong inhibitory signal, as deficiency in Ly108 is sufficient to alleviate the SAP requirement for the GC reaction (*Kageyama et al., 2012*). *Itsn2⁻/⁻* GC B cells express reduced levels of SLAM and CD84, but not of Ly108, which could alter the signalling mediated by these receptors, and hence the outcome of B-T interactions.

In summary, we have delineated a role for ITSN2, an interacting partner of Cdc42, WASp, and WIP, in generating optimal immune responses. We probed the function of this protein in the B cell compartment by studying ITSN2 deficient mice and mixed BM chimeras and using a combination of immunisation and infection experiments. Our results identify ITSN2 as a regulator of B cell activation, engagement into the GC reaction and cognate B-T interactions. They open new perspectives to understand the intricate regulatory network underlying cognate interactions between B and T cells.

# Materials and methods

## Key resources table

| Reagent type (species) or resource | Designation | Source or reference | Identifiers | Additional information |
|---|---|---|---|---|
| Gene (*Mus musculus*) | *Itsn2* | NA | ENSMUSG00000020640; MGI:1338049 | |
| Genetic reagent (*M. musculus*) | *Itsn2⁻/⁻* | KOMP consortium | RRID:MGI:5631233 | Itsn2^tm1.1(KOMP)Vlcg |
| Genetic reagent (*M. musculus*) | μMT | doi:10.1038/350423a0 | RRID:IMSR_HAR:1682 | (beta)-globin (mu)MT⁻/⁻ |
| Genetic reagent (*M. musculus*) | MD4 | doi:10.1038/334676a0 | RRID:MGI:5006966 | Tg(IghelMD4)4Ccg |
| Genetic reagent (*M. musculus*) | OTII | doi:10.1046/j.1440–1711.1998.00709.x | RRID:IMSR_JAX:004194 | B6.Cg-Tg(TcraTcrb)425Cbn/J |
| Cell line (*M. musculus*) | A20 cells with HEL-specific D1.3 BCR | http://dx.doi.org/10.1016/S1074-7613(00)80580–4 | | |
| Transfected construct (plasmid) | ITSN2L GFP | Michael Way | | pEGFP-C1 ITSN2L |
| Antibody (rabbit polyclonal) | anti-ITSN2 antibody | Novus | NBP1-71833; RRID:AB_11038593 | 1:1000 in 5% Milk TBS-T |
| Sequence-based reagent | sgRNA1 | this paper | | Forward - CACCGTAGCTATAGAGAACTCTTGC; Reverse - AAACGCAAGAGTTCTCTATAGCTAC |
| Sequence-based reagent | sgRNA 2 | this paper | | Forward - CACCGGGGGGTTGTTTCATGATAGGA; Reverse - AAACTCCTATCATGAAACAACCCCC |

*Continued on next page*

*Continued*

| Reagent type (species) or resource | Designation | Source or reference | Identifiers | Additional information |
|---|---|---|---|---|
| Peptide, recombinant protein | Eα peptide | The Francis Crick Institute peptide chemistry unit; doi: 10.1038/353660a0 | | Sequence: Biotin-GSGFAKFASFEAQGA LANIAVDKA-COOH |

## Animal breeding and generation

ITSN2<sup>tm1.1Komp(Vlcg)</sup> (RRID:MGI:5631233) animals, where ITSN2 expression is disrupted by the insertion of a LacZ cassette were obtained from the KOMP consortium (*Skarnes et al., 2011*; *Valenzuela et al., 2003*). They were crossed to mice expressing the hen egg lysozyme (HEL) specific MD4 B cell receptor (RRID:MGI:5006966, [*Goodnow et al., 1988*]). OTII mice were used for purification of ovalbumin (OVA) restricted CD4 T cells (RRID:IMSR_JAX:004194, [*Barnden et al., 1998*]). For adoptive transfer experiments, congenic CD45.1 C57BL6 mice were obtained from the internal breeding facility at the Francis Crick Institute and µMT ([*Kitamura et al., 1991*], RRID:IMSR_HAR: 1682, B cell deficient) mice were bred internally. C57BL6/J mice were either obtained internally or purchased from the Jackson Laboratory.

For the generation of mixed bone marrow chimeras, 6–8 week-old recipient mice were provided acidified water one week before the beginning of the procedure, lethally irradiated using $2 \times 6$ Gy, and intravenously injected 24 hr later with $2.10^6$ donor bone marrow cells in adequate proportions. Animals were bled 8–10 weeks after adoptive transfer to check the reconstitution efficiency, and were subsequently used for experiments.

Mice were bred and maintained at the animal facility at the Francis Crick Institute. All experiments were approved by the Animal Ethics Committee of the Francis Crick Institute and the UK Home Office and by the Institutional Animal Care and Use Committee (IACUC) of the Massachusetts General Hospital. Animal experiments conducted at the Ragon Institute were performed in accordance with the regulations of the American Association for the Accreditation of Laboratory Animal Care (AAALAC).

## Cell isolation, labelling and culture

Splenic naive B or CD4 T lymphocytes were purified using negative B cell or CD4 T cell isolation kits yielding enriched populations of 95–98% (B cells) and 80% (T cells), respectively (Miltenyi Biotec, Germany). Purified B or T cells were labelled in PBS with 2 µM CTV (Life Technologies, Carlsbad, California), 2 µM SNARF1 (Invitrogen), or 1 µM CFSE (Life Technologies) for 5 min at 37°C and washed (twice in case of CFSE labelling) in complete B cell medium. Cells were maintained in complete B cell medium (RPMI supplemented with 10% FCS, 25 mM Hepes, Glutamax, penicillin streptomycin (Invitrogen), and 100 µM β-mercaptoethanol (Sigma-Aldrich, St. Louis, Missouri)).

## Cell lines and transient transfection

A20 lymphoma cells expressing HEL-specific D1.3 receptor (*Batista and Neuberger, 1998*) generated in the laboratory were cultured in RPMI with 10% FCS, 25 mM HEPES, glutamax, penicillin streptomycin (Invitrogen) and 100 µM β-mercaptoethanol. Cells were transfected using Amaxa Nucleofactor Technology (KitV, programme L-013, Lonza, Switzerland) according to the manufacturer's instructions. Mycoplasma contamination status was not tested. Transfected cells were used within 5–8 hr after transfection. Constructs encoding GFP-ITSN2-L were kindly provided by M.Way (The Francis Crick Institute).

## CRISPR/Cas9-mediated ITSN2 deletion

ITSN2 was deleted in A20 cells using the CRISPR/Cas9 system. Briefly, sgRNA targeting ITSN2 were cloned in the vectors expressing Cas9GFP (Adgene PX458); and the following targeting regions were used: CACCGTAGCTATAGAGAACTCTTGC and AAACGCAAGAGTTCTCTATAGCTAC (sgRNA1) or CACCGGGGGGTTGTTTCATGATAGGA and AAACTCCTATCATGAAACAACCCCC (sgRNA2). The constructs were transiently expressed in derivatives of A20 B cell line expressing hen egg lysozyme (HEL)-specific D1.3 BCR for 12 hr followed by single-cell sorting of GFP-positive cells into 96-well plates. The cells were maintained in complete B cell medium. Single cell clones were

expanded and screened for ITSN2 deletion by immunoblotting. Clones were matched to WT cells for receptor expression and clones obtained with both sgRNAs were retained for experiments.

## Immunisation, ELISA and ELISPOT

For immunization, mice were injected i.p. with 50 µg NP-KLH (Biosearch Technology, Novato, California) in 4 mg Alum (Thermo Fisher Scientific, Waltham, Massachussets). Blood samples were taken from the lateral tail vein on 0, 7, 14, 21, 28 days after immunization.

NP-specific antibody titres were detected by ELISA, using NP20-BSA or NP4-BSA for capture, and biotinylated anti–mouse IgM (Southern Biotech, Birmingham, Alabama), IgG (Southern Biotech), IgG2b (Southern Biotech), IgG2c (Southern Biotech), and IgG3 (BD, Franklin Lakes, New Jersey) for detection. Titres were determined from the dilution curve in the linear range of absorbance.

For detection of NP-specific antibody secreting cells, Multiscreen filter plates (Millipore, Burlington, Massachussets) were activated for 1 min with ethanol (Sigma Aldrich), washed with sterile PBS, and coated with 1 µg.mL$^{-1}$ NP20-BSA in PBS overnight at 4°C. After blocking in complete B cell medium, splenocytes collected from immunised mice were plated at various concentrations. Biotinylated anti-mouse IgM or IgG (Southern Biotech) were used for detection.

ELISA and ELISPOT plates were developed with alkaline-phosphatase streptavidin (Sigma-Aldrich) and phosphorylated nitrophenyl phosphate (ELISA, Sigma-Aldrich) or BCIP/NIST (ELISPOT, Sigma-Aldrich). Absorbance at 405 nm was determined with a SPECTRA-max190 plate reader (Molecular Devices, Sunnyvale, California). Imaging of ELISPOT plates was performed using an Immunospot analyser (Immunospot, Cleveland, Ohio).

## Infection

For infection, mice were intranasally immunized with $3.10^4$ pfu of Influenza A virus (PR8 strain) or injected with $10^4$ pfu of Vaccinia virus (Western Reserve) intra-footpad. For survival experiments, animals were weighed daily and sacrificed when exhibiting 20% wt loss. For Influenza vaccination, animals were immunised with hemagglutinin trimer (kindly provided by Daniel Lingwood) together with Sigma adjuvant by intravenous route, and 4 weeks later challenged with $1.5.10^5$ pfu Influenza A PR8. Animals were weighed daily, and sacrificed when exhibiting 40% wt loss. In the case of infection with Vaccinia virus, immune responses in the draining lymph nodes were characterised 7 days after infection.

## Flow cytometry

Single cell suspensions were prepared from spleen, lymph nodes, bone marrow or peritoneal lavage. After blocking Fc receptors using anti CD16/32 antibodies, cells were stained with the appropriate combination of the following antibodies: B220 (RA3-6B2), TCRβ (H57-597), CD21 (7G6), CD4 (GK1.5), CD8 (53–6.7), CD3 (145–2 C11), CD23 (B3B4), CD24 (M1/69), CD43 (S7), Ly-51 (6C3), B and T cell activation antigen (GL7), CD95 (Jo2), CD138 (281.2), IgG1 (A85.1), CD5 (53–7.3), CD11b (M1/70), CD19 (1D3), IgM (1B4B1 or II/41), IgD (11-26), CXCR4 (2B11), CD86 (GL1), CD84 (mCD84.7), ICOSL (HK5.3), SLAM (TC15-T2F12.2), Ly108 (330-A5), CXCR5 (2G8), PD-1 (543), CD45.1 (A20), CD45.2 (104), MHCII (25-9-17), CD40 (3/23), CD69 (H1-2F3).

Data was acquired on LSR Fortessa (BD Bioscience) and analyzed with FlowJo (Treestar, Ashland, Oregon).

## Protein biotinylation

Ovalbumin (OVA, Calbiochem) or hen egg lysozyme (HEL, Sigma-Aldrich) were used at a concentration of 1 mg.mL-1 in PBS and mixed for 30 min at room temperature with EZ Link-Sulfo-NHS-Biotin reagent (Pierce). For monobiotinylation, typical concentrations of 0.01 µg.mL-1, 0.02 µg.mL-1, 0.05 µg.mL-1. For poly-biotinylation, higher concentrations, ranging from 0.5 to 1 µg.mL-1 were used. Excess free NHS-biotin was excluded by dialysing the mixtures in PBS overnight at 4°C.

The degree of biotinylation was tested by flow cytometry. To this end, dialysed antibodies were incubated for 20 min at room temperature with 5.6 µm streptavidin polystyrene beads. Known mono- and poly-biotinylated antibodies were used as negative and positive controls, respectively. Antibodies coupled to beads were detected using FITC-conjugated anti-rat secondary antibodies,

and additional free biotin sites were detected using Streptavidin Alexa Fluor 633. Beads were washed with PBS and analysed by flow cytometry.

## Beads preparation

Unlabelled (0.11 µM diameter) or AlexaFluor 647 (0.22 µM diameter) streptavidin coated microspheres (Bangs laboratories, Fishers, Indiana) were incubated with saturating amount (as determined from manufacturer's instructions) of biotinylated anti-IgM (Southern Biotech) or HEL and Ovalbumin (Calbiochem) or Eα peptide for 1 hr at 37°C. Limiting stimulatory conditions were obtained by increasing the amounts of Ova or Eα peptide for coating while IgM amounts were kept constant. Efficient titration of the IgM signal was measured by flow cytometry. For ex vivo internalisation assays, beads coated with anti-IgM alone were used.

## In vivo B cell activation

WT and $Itsn2^{-/-}$ B cells labelled in CFSE or CTV (colour-switched experiments were performed systematically) were transferred to C57BL6 (for multiphoton and localisation analysis) or C57BL/6. CD45.1 (for in vivo proliferation and short activation experiments) recipients together with OTII T cells (labelled in CFSE or SNARF-1 when specified) in a typical ratio of $2.10^6$ B cells for $10^6$ T cells. 24 hr after adoptive transfer, recipient animals were immunised with beads coated with HEL and Ovalbumin, either via tail vein injection (for in vivo proliferation and short-term activation experiments), or by intra-footpad injections (for multiphoton and localisation analysis).

## Antigen internalization and presentation

For internalization assays, purified B cells were loaded with IgM coated fluorescently-labelled microspheres or with soluble biotinylated anti-IgM (Southern Biotech) on ice for 30 min. Cells were then washed with PBS 2% FCS to remove excess antigen and then incubated for 5 to 30 min at 37°C. After fixation with 2% paraformaldehyde, non-internalised beads or anti-IgM were detected with AlexaFluor450 streptavidin (eBioscience, San Diego, California). In case of bead internalisation, bead-positive cells were selected on the basis of the bead fluorescence.

To detect antigen presentation, B cells loaded with Eα peptide and IgM coated microspheres were incubated between 3 and 5 hr at 37°C and then fixed in 2% formaldehyde. These cells were then stained with anti-MHCII/Eα antibody (eBioY-Ae, eBioscience), followed by anti-mouse IgG2b antibody staining (Life Technologies) for detection.

## Proliferation analysis

CFSE or CTV-labelled cells at a concentration of $10^6$ cells per mL were stimulated in complete B cell medium supplemented with combinations of 1–10 µg.mL$^{-1}$ anti-IgM F(ab')$_2$ (Jackson Immunoresearch, West Grove, Pennsylvania), 1–10 µg.mL$^{-1}$ of LPS (Sigma), 0.3–1 µg.mL$^{-1}$ CD40L (R and D systems, Minneapolis, Minnesota) or 0.3–3 µg.mL$^{-1}$ CpG (ODN1826, Sigma), 10 ng.mL$^{-1}$ of IL4 (R and D systems) or 10 ng.mL$^{-1}$ of IL5 (R and D systems). CFSE or CTV dilution was measured after 3 or 4 days by flow cytometry.

For B-T coculture, CTV-labelled B cells were loaded with beads coated with IgM and OVA for 30 min at 37°C in complete medium, then washed with complete medium to remove excess microspheres and subsequently cultured with CFSE-labelled OTII T cells in a 1:1 ratio. B and T cell proliferation were measured after 3 days by flow cytometry.

## Immunoprecipitation and western blotting

Purified B cells were left at 37°C for 10 min in chamber buffer (PBS, 0.5% FCS, 1 g.L$^{-1}$ D-Glucose, 2 mM MgCl2, 0.5 mM CaCl2) to equilibrate prior to stimulation. They were then stimulated for various times with 10 µg.mL$^{-1}$ anti-IgM F(ab')$_2$ fragment (Jackson Immunoresearch), prior to lysis in protein lysis buffer (20 mM Tris-HCL, pH 8, 150 mM NaCl, 5 mM EDTA, Protease Inhibitor cocktail (Roche, Switzerland), Phosphatase inhibitors (10 mM NaF, 1 mM Na$_3$VO$_4$), 1% NP40) for 30 min on ice. Lysates were subsequently centrifuged for 15 min at maximum speed on a tabletop centrifuge, and post-nuclear supernatants used for further analysis.

For immunoprecipitation, post-nuclear supernatants were immunoprecipitated with the 4G10 Ab (Millipore) prebound to protein G-Sepharose beads (Sigma). After gentle rotation at 4°C for 2 hr,

the beads were washed five times with protein lysis buffer, and bound proteins were eluted with Laemmli buffer.

After addition of 2X Laemmli buffer and denaturation at 95°C for 5 min, samples were resolved by SDS-PAGE on precast minigels (7.5% or 12% Tris-HCl; Bio-Rad, Hercules, California), and transferred electrophoretically to PVDF filters using the Miniprotean Transblot system (Bio-Rad). Membranes were incubated in WB blocking buffer (5% milk in TBS, 0.1% Tween) to block non-specific binding, and probed with specific Abs diluted in WB blocking buffer or 5% BSA in TBS-T. Primary antibodies used were specific for ITSN2 (Novus, St-Louis, Missouri, RRID:AB_11038593), α-tubulin (Sigma), ERK (Cell Signaling Technology, RRID:AB_330744), phospho-ERK (Cell Signaling Technology, Danvers, Massachussets, RRID:AB_331646), phospho-Akt-S (Cell Signaling Technology, RRID:AB_329825), phospho-SHP1 (Cell Signaling Technology, RRID:AB_11141050), phospho-CD19 (Cell Signaling Technology, RRID:AB_2072836). After extensive washing in TBS-T, the membranes were incubated with HRP-labeled goat anti-mouse Ig or goat anti-rabbit Ig Abs (Jackson Immunoresearch), and immunoreactivity was visualised by using the ECL system (Amersham Biosciences).

Densitometric analysis of the films was performed using the Image J software.

## Optical microscopy

Planar lipid bilayers containing anti-mouse κ light chain (HB-58; American Type Culture Collection) were prepared in FCS2 chambers (Bioptechs Inc., Butler, Pennsylvania) by liposome spreading as previously described (Carrasco et al., 2004). Briefly, Alexa-633 streptavidin (Molecular Probes, Eugene, Oregon) was incorporated into lipid bilayers at density of 30 molecules/μm$^2$, to which mono-biotinylated antigen was tethered. Assays were performed in chamber buffer at 37°C and imaged with TIRFM.

For spreading analysis, cells were settled on coverslips coated either with anti-κ chain antibody (HB-58; American Type Culture Collection), or anti MHCII antibody (TIB120; American Type Culture Collection) at 37°C. After 10 min, cells were fixed with prewarmed 2% paraformaldehyde, permeabilised with PBS 0.1% Triton for 1 min and stained with Alexa Fluor 488 phalloidin (Molecular Probes).

TIRF images were acquired with an EMCCD camera (iXon3 897, Andor, Northern Ireland) coupled to a TIRF microscopy system (Cell R; Olympus, Japan) with 488 nm, 561 nm and 640 nm lasers (Olympus). Images were recorded with Cell R software (Olympus) and analyzed with ImageJ software (NIH).

Lymph nodes and spleen were embedded in OCT and frozen in cold isopentane. 10 μm wide frozen sections were cut with a cryostat. Sections were fixed in 4% paraformaldehyde, blocked with PBS containing 1% BSA and 10% goat serum (IF blocking buffer). Staining was performed in IF blocking buffer with a combination of the following antibodies: GL7-AF647, B220 PB (RA3-6B2), TCRβ FITC or PE (H57-597), anti-κ PE (187.1), F4/80 APC (BM8). Confocal imaging was performed with a Zeiss LSM 780 microscope, using a 20x objective (Plan-Appochromat 20X/0.8NA) for spleen sections or a 40x objective (Plan-Appochromat 20X/1.3NA, oil immersion) for LN sections. Tiled images were stitched using the ZEN software (Zeiss), and analysed in Imaris (Bitplane) or ImageJ (NIH).

For multiphoton microscopy, explanted popliteal and inguinal lymph nodes were prepared as described (Carrasco and Batista, 2007), and imaged with an upright multiphoton microscope (Olympus), a 25X, NA 1.05 water immersion objective, and a pulsed Ti:sapphire laser (Spectra Physics MaiTai HP DeepSee) tuned to 820 nm. Emission wavelengths were detected through band-pass filters of 420–500 nm (CTV), 515–560 nm (CFSE) and 590–650 nm (SNARF-1). Multidimensional movies were analyzed with Imaris. Fluorescence bleed-through into the channel with longer wavelength was removed by subtracting the channel with shorter wavelength. Interaction times were measured using spot colocalization with a distance threshold of 10 μm, using Matlab (The MathWorks, Natick, Massachussets).

## Experimental data and statistical analysis

Sample sizes were chosen on the basis of published work in which similar phenotypical characterization and similar defects were reported. Cohort randomization or 'blinding' of investigators to sample identity was not done in this study. In some experiments, the data for each study group were compared with Student's t-test or (when applicable) with a paired t-test and P values were calculated. Normal distribution of samples was assumed on the basis of published studies with analysis similar

to ours. All statistical analysis was performed using the Prism software (Graphpad Inc, San Diego, California).

## Acknowledgements

We thank the biological resource unit for animal husbandry (The Francis Crick Institute), the flow cytometry unit (The Francis Crick Institute), and the peptide chemistry unit for synthesis of Eαpeptide (The Francis Crick Institute). The *Itsn2*$^{-/-}$ strain was generated by the trans-NIH Knock-Out Mouse Project (KOMP) and obtained from the KOMP Repository (www.komp.org). We thank Daniel Lingwood for kindly sharing with us the Hemagglutinin trimer and Flu vaccination protocol. We thank Beatriz Montaner and Kathrin Kirsch for help with managing animal colonies. We thank Selina J Keppler and all the members of the Lymphocyte Interaction Laboratory for scientific discussions and critical reading of the manuscript. We thank Jan Sondenkamp (The Francis Crick Institute) for imaging the ELISPOT plates. This work was supported by the Francis Crick Institute which receives its core funding from Cancer Research UK (FC001035, FC001209), the UK Medical Research Council (FC001035, FC001209), and the Wellcome Trust (FC001035, FC001209), The Center for HIV/AIDS Vaccine Immunology and Immunogen Discovery of the National Institutes of Health (UM1AI100663), the Phillip T and Susan M Ragon Institute Foundation, a Institute Pasteur–Fondazione Cenci Bolognetti (FG) and a long term fellowship from EMBO (SA).

## Additional information

### Competing interests

Facundo D Batista: Reviewing editor, *eLife*. The other authors declare that no competing interests exist.

### Funding

| Funder | Grant reference number | Author |
|---|---|---|
| Cancer Research UK | FC001035 | Marianne Burbage<br>Francesca Gasparrini<br>Shweta Aggarwal<br>Andreas Bruckbauer<br>Facundo D Batista<br>Mauro Gaya |
| Medical Research Council | FC001035 | Marianne Burbage<br>Francesca Gasparrini<br>Shweta Aggarwal<br>Andreas Bruckbauer<br>Facundo D Batista<br>Mauro Gaya |
| Wellcome | FC001035 | Marianne Burbage<br>Francesca Gasparrini<br>Shweta Aggarwal<br>Andreas Bruckbauer<br>Facundo D Batista<br>Mauro Gaya |
| Institut Pasteur | Fondazione Cenci Bolognetti | Francesca Gasparrini |
| EMBO | ALTF 302-2013 | Shweta Aggarwal |
| Medical Research Council | FC001209 | Michael Way |
| Wellcome | FC001209 | Michael Way |
| Cancer Research UK | FC001209 | Michael Way |
| The Center for HIV/AIDS Vaccine Immunology and Immunogen Discovery of the National Institutes of Health | UM1AI100663 | Facundo D Batista |

The funders had no role in study design, data collection and interpretation, or the decision to submit the work for publication.

## Author contributions
Marianne Burbage, Investigation, Acquisition of data, Writing—original draft, Conception and design, Analysis and interpretation of data, Drafting or revising the article; Francesca Gasparrini, Shweta Aggarwal, Mauro Gaya, Acquisition of data, Analysis and interpretation of data, Editing the manuscript; Johan Arnold, Acquisition of data, Analysis and interpretation of data; Usha Nair, Drafting or revising the manuscript; Michael Way, Resources, Writing—review and editing; Andreas Bruckbauer, Acquisition of imaging data and image analysis, editing the manuscript; Facundo D Batista, Investigation, Writing—original draft, Conception and design, Analysis and interpretation of data, Drafting or revising the article

## Author ORCIDs
Marianne Burbage ⓘ http://orcid.org/0000-0003-0686-2608
Facundo D Batista ⓘ https://orcid.org/0000-0002-1130-9463

## Ethics
Animal experimentation: All experiments involving animals were approved by the Animal Ethics Committee of The Francis Crick Institute and the UK Home Office (Project licence n°70/8683), and by the Institutional Animal Care and Use Committee of the United States (2016N00286).

## Decision letter and Author response
Decision letter https://doi.org/10.7554/eLife.26556.022
Author response https://doi.org/10.7554/eLife.26556.023

# Additional files

## Supplementary files
• Transparent reporting form
DOI: https://doi.org/10.7554/eLife.26556.018

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
