## [Decision Letter]

Thank you for submitting your article "Tuning of in vivo cognate B-T cell interactions by Intersectin 2 is required for effective anti-viral B cell immunity" for consideration by *eLife*. Your article has been reviewed by two peer reviewers, one of whom, Michael L Dustin is a member of our Board of Reviewing Editors and the evaluation has been overseen by Michel Nussenzweig as the Senior Editor.

The reviewers have discussed the reviews with one another and the Reviewing Editor has drafted this decision to help you prepare a revised submission.

Summary:

In this study, Burbage and colleagues explore the role of ITSN2, an SH3- containing adaptor protein and a Cdc42 GEF that is associated with Sjogren's syndrome, in B cell activation, germinal center development, and antibody responses against model antigen and viral infection. The study is quite comprehensive and touches on multiple potentially interesting and important topics of the field (e.g. a new player mediating specialized signals downstream of BCR for activation and/or antigen uptake; potential involvement of B-cell cytoskeleton dynamics in regulation of antigen-specific T:B interactions). However, despite a multitude of phenotypes observed in association with the ITSN2 deficiency, mechanistic analyses remain superficial and fragmented. In fact, different parts of the study are not so well connected to one another, and as a whole the study cannot yet reach a firm conclusion on the role of ITSN2 at the cellular, tissue, or the organismal level.

Essential revisions:

1) Although ITSN2 KO mice show an increased susceptibility to PR8 infection within the first 3-6 days, it is too acute to be explained by potentially abnormal B-cell activation or defective germinal center development (observed on day 7 in Figure 2). The readers are left wondering exactly how this phenotype is connected to later parts of the paper, particularly those mechanistic studies. The reviewers could envision two ways to address this gap. (1) its known that natural antibodies and complement play a role in protection of naive mice against flu infection (see Jayasekera, Moseman and Carroll, 2007). The authors surprisingly don't report on basal serum antibodies and ability of these to neutralize the PR8 virus in vitro. If this information is provided and accounts for the early defect then this could be reported as an additional defect that would connect the resistance to flu to a B cell defect and would be sufficient to go forward with studies of vaccination responses. (2) Since most of the study focuses on germinal center and recall responses, could the flu infection be done in the context of vaccination looking for lack of or partly defective protection to re-infection? This could also address the gap, although perhaps the data could be presented after the analysis of GC reactions.

2) These authors suspected antigen uptake/processing might be impaired in the absence of ITSN2, and three assays in vitro, namely, BCR internalization, direct detection of surface pMHC complexes, and T cell-driven B cell proliferation in culture to test if ITSN2 is required for efficient BCR-mediated antigen uptake and presentation (Figure 7—figure supplement 1). These are important and appropriate assays, especially given the subsequent observation of defective T:B interactions in vivo (Figure 8). However, it is VERY difficult for these in vitro assays to mimic the level of antigen availability and the stringent requirement for cognate T cells that antigen-specific B cells may experience in vivo. The fact that an effect of ITSN2 was not apparent in those in vitro assays may still reflect a sensitivity issue. They should try to test BCR internalization and pMHC levels on WT and ITSN2 KO B cells activated in vivo as in Figure 7. Also, they should include a group without T cell co-transfer in Figure 7 to ensure the difference in proliferation seen is T cell-dependent. It is really hard to conceptualize that differences in CD84, SLAM, and ICOSL as seen in Figure 7 can explain the defect in proliferation. It remains the more likely scenario that the ITSN2 deficiency impairs BCR signaling and antigen uptake, leading to defects in antigen-specific T:B interactions. The observation that WT and KO B cells respond to NP-LPS differently (Figure 4) also supports the possibility that defects in BCR and/or TLR signaling is a primary defect. The authors should look harder.

3) Analysis of the ITSN2 impact on BCR signaling is somewhat superficial, and the authors did not offer an explanation for the difference between reduced CD19/PI3K activities and enhanced ERK activation (Figure 6). What about calcium flux after anti-IgM stimulation? Are any of these pathways more or less important for BCR-mediated antigen uptake and CD84/SLAM/ICOSL expression?

4) The LN explant imaging data in Figure 8 look solid, although it is important to conduct color-switched experiments to ensure the difference seen is not due to different toxicity profiles of the organic fluorescence dyes?

5) It is probably important to conduct rescue experiments with ITSN2 overexpression in some of the relevant assays. This is to ensure the effect seen with the ITSN2 deficiency is indeed due to ITSN2 but not LacZ, given how the KOMP knockout was made.

---

## [Author Response]

Essential revisions:1) Although ITSN2 KO mice show an increased susceptibility to PR8 infection within the first 3-6 days, it is too acute to be explained by potentially abnormal B-cell activation or defective germinal center development (observed on day 7 in Figure 2). The readers are left wondering exactly how this phenotype is connected to later parts of the paper, particularly those mechanistic studies. The reviewers could envision two ways to address this gap. (1) its known that natural antibodies and complement play a role in protection of naive mice against flu infection (see Jayasekera, Moseman and Carroll, 2007). The authors surprisingly don't report on basal serum antibodies and ability of these to neutralize the PR8 virus in vitro. If this information is provided and accounts for the early defect then this could be reported as an additional defect that would connect the resistance to flu to a B cell defect and would be sufficient to go forward with studies of vaccination responses. (2) Since most of the study focuses on germinal center and recall responses, could the flu infection be done in the context of vaccination looking for lack of or partly defective protection to re-infection? This could also address the gap, although perhaps the data could be presented after the analysis of GC reactions.

We thank the reviewers for highlighting this omission. We have now measured basal serum antibody titres, and found no notable differences between WT and ITSN2 KO animals. This is now shown in Figure 1—figure supplement 1.

Our revised version of the manuscript also describes flu infection in the context of vaccination. To address this, we applied a strategy described by Zheng et al., (2016), wherein animals are vaccinated by intravenous injection of hemagglutinin trimer together with Sigma Adjuvant. After 1 month, animals are injected with a lethal dose (1.5x10^5)^of PR8 virus. In response to such challenge, both WT and ITSN2 KO mice exhibited severe weight loss. As expected from this experiment, WT animals started to regain weight at d9 post infection; however, this was not the case for the ITSN2 KO animals, whose weight continued to remain low for the entire duration of the experiment. These results further support the notion that ITSN2 is required for establishing protective responses upon viral infection. They are now presented in Figure 2. We believe that these results strengthen our study, and thank the reviewers for highlighting this important point.

2) These authors suspected antigen uptake/processing might be impaired in the absence of ITSN2, and three assays in vitro, namely, BCR internalization, direct detection of surface pMHC complexes, and T cell-driven B cell proliferation in culture to test if ITSN2 is required for efficient BCR-mediated antigen uptake and presentation (Figure 7—figure supplement 1). These are important and appropriate assays, especially given the subsequent observation of defective T:B interactions in vivo (Figure 8). However, it is VERY difficult for these in vitro assays to mimic the level of antigen availability and the stringent requirement for cognate T cells that antigen-specific B cells may experience in vivo. The fact that an effect of ITSN2 was not apparent in those in vitro assays may still reflect a sensitivity issue. They should try to test BCR internalization and pMHC levels on WT and ITSN2 KO B cells activated in vivo as in Figure 7.

We thank the reviewers for pointing out these important points. We have indeed measured the presentation of Eα peptide not only in vitro (Figure 7—figure supplement 1A), but also in vivo (Figure 7). We have now amended the text and figure legends to state this more explicitly. Moreover, to complement these results, we have now included information regarding the ability of ITSN2 KO B cells to capture particulate antigen in vivo. These data are now presented in (Figure 6—figure supplement 1, panel F) and discussed in the text.

Also, they should include a group without T cell co-transfer in Figure 7 to ensure the difference in proliferation seen is T cell-dependent.

in vivo proliferation using Hel-OVA coated beads has been extensively used in our group. To confirm that MD4^+^ B cell proliferation is T cell-dependent in this setting, we have performed multiple experiments comparing beads coated with HEL in absence of OVA. Under these conditions, we were unable to detect B cell proliferation, indicating that B cell proliferation is dependent on T cell help. We have attached for the reviewer’s attention, representative results of such experiments (Author response image 1), and would be happy to include them as supplementary figures if they feel it would be useful for the clarity of the manuscript.

**Author response image 1. respfig1:** MD4 B cells and OTII T cells were transferred to WT recipients, and animals immunised with particles coated with HEL alone (black tinted line) or HEL and OVA (Blue line). B cells proliferation was measured by FACS at d3.

It is really hard to conceptualize that differences in CD84, SLAM, and ICOSL as seen in Figure 7 can explain the defect in proliferation. It remains the more likely scenario that the ITSN2 deficiency impairs BCR signaling and antigen uptake, leading to defects in antigen-specific T:B interactions. The observation that WT and KO B cells respond to NP-LPS differently (Figure 4) also supports the possibility that defects in BCR and/or TLR signaling is a primary defect. The authors should look harder.

We thank the reviewers for these comments. We have now analysed B cell proliferation in response to particles coated with IgM and CpG in vitro, and found no differences in the ability of ITSN2 KO and WT B cells to respond to such stimuli. This is now reported in Figure 5—figure supplement 2,panel E.

Moreover, we have also explored the long-term consequences of BCR triggering by stimulating WT and ITSN2 KO B cells in vitro in presence of IL4 with or without anti-IgM for 24h. In these settings, we found that ITSN2 KO B cells presented consistently reduced levels of phospho-SHP1, but not phosphor-ERK in presence, but not in absence of anti-IgM. This is now reported in Figure 5. These results suggest that ITSN2 deficiency causes long-term differences in BCR signaling.

3) Analysis of the ITSN2 impact on BCR signaling is somewhat superficial, and the authors did not offer an explanation for the difference between reduced CD19/PI3K activities and enhanced ERK activation (Figure 6). What about calcium flux after anti-IgM stimulation?

We thank the reviewers for their suggestions. We have now analysed calcium signalling upon anti-κ stimulation, and found that calcium flux upon BCR triggering was not impaired in ITSN2 deficient cells. This is now reported in Figure 5 and discussed in the text.

We recently described that B cells lacking the WASp interacting protein (WASp) also have defective GC formation and antibody responses, and present with altered PI3K signalling and CD19 phosphorylation, but not ERK activation, downstream BCR engagement, associated with altered BCR motility. To test whether this association could be observed in ITSN2 KO cells, we compared the diffusion coefficient of IgM, IgD and CD19 on WT and ITSN2 KO B cells. We detected a mild increase in CD19 diffusion coefficient in ITSN2 KO B cells, concordant with the model we proposed wherein WIP regulated CD19 phosphorylation by modulating actin dynamics and BCR motility.

Are any of these pathways more or less important for BCR-mediated antigen uptake and CD84/SLAM/ICOSL expression?

Expression of CD84, SLAM and ICOSL is increased in B cells upon activation. Although we could not detect B cell activation defects in vitro upon ITSN2 deletion, we cannot exclude that in vivo, the reduced expression of these receptors on activated ITSN2 KO B cells results from BCR signalling defects.

4) The LN explant imaging data in Figure 8 look solid, although it is important to conduct color-switched experiments to ensure the difference seen is not due to different toxicity profiles of the organic fluorescence dyes?

We thank the reviewers for pointing this out. We have indeed performed color-switched experiments to rule out dye-specific effects and have found similar results. Colour-switched experiments were also performed for all adoptive transfer experiments where WT and ITSN2 KO cells were analysed in the same animal. We now have clarified the legend and material and methods to reflect this important point.

5) It is probably important to conduct rescue experiments with ITSN2 overexpression in some of the relevant assays. This is to ensure the effect seen with the ITSN2 deficiency is indeed due to ITSN2 but not LacZ, given how the KOMP knockout was made.

We thank the reviewers for highlighting this. Upon reception of the ITSN2 KO strain, we performed experiments (development and immunization) comparing WT, Heterozygous and ITSN2 KO animals. We never detected any difference between WT and heterozygous animals, and hence have used heterozygous animals (that express one copy of lacz) as controls in all experiments presented in the paper. We present here (Author response image 2) an experiment where ITSN2+/- animals were compared to WT ones in the context of Vaccinia infection, showing germinal centres in the popliteal LN.

**Author response image 2. respfig2:** WT and ITSN2+/- littermates were infected with 10^4^ PFU VACV by intra-footpad injection. GC formation was measured in the popliteal LN 7 days after infection.

Moreover, to address this point, we have used CRISPR/Cas9 to ablate ITSN2

in A20 cells expressing a HEL-specific BCR. In this system, independent of

LacZ knock-in, we could reproduce the defect in actin foci formation observed

in primary ITSN2 KO cells. Moreover, complementation of the ITSN2 KO cells

with ITSN2LGFP was sufficient to restore actin foci formation to the WT level.

This is now reported in Figure 5—figure supplement 1